# Inference of germinal center evolutionary dynamics via simulation-based deep learning

Duncan K Ralph[1]*, Athanasios G Bakis[2], Jared G Galloway[1], Ashni A Vora[3], Tatsuya Araki[3], Gabriel D Victora[3,4], Yun S Song[5,6], William S DeWitt[1,7], Frederick A Matsen[1,4,7,8]*

[1]Fred Hutchinson Cancer Research Center, Seattle, United States; [2]Department of Statistics, University of California, Irvine, Irvine, United States; [3]Laboratory of Lymphocyte Dynamics, The Rockefeller University, New York, United States; [4]Howard Hughes Medical Institute, Seattle, United States; [5]Department of Statistics, University of California, Berkeley, Berkeley, United States; [6]Computer Science Division, University of California, Berkeley, Berkeley, United States; [7]Department of Genome Sciences, University of Washington, Seattle, United States; [8]Department of Statistics, University of Washington, Seattle, United States

*For correspondence:
dralph@fredhutch.org (DKR);
matsen@fredhutch.org (FAM)

## eLife Assessment

This paper presents a computational method to infer from data a key feature of affinity maturation: the relationship between the affinity of B-cell receptors and their fitness. The approach, which is based on a simple population dynamics model but inferred using AI-powered simulation-based inference, is novel and **valuable**. It exploits recently published data on replay experiments of affinity maturation. The method is well argued and presented, and the validation is **compelling**.

**Abstract** B cells and the antibodies they produce are vital to health and survival, motivating research on the details of the mutational and evolutionary processes in the germinal centers (GCs) from which mature B cells arise. It is known that B cells with higher affinity for their cognate antigen (Ag) will, on average, tend to have more offspring. However, the exact form of this relationship between affinity and fecundity, which we call the 'affinity–fitness response function', is not known. Here we use deep learning and simulation-based inference to learn this function from a unique experiment that replays a particular combination of GC conditions many times in mice. All code is freely available at https://github.com/matsengrp/gcdyn, while datasets and inference results can be found at https://doi.org/10.5281/zenodo.15022130.

## Introduction

The germinal center (GC) is the site of affinity maturation of B cell receptors (BCRs), and as such is of central importance to the functioning of the adaptive immune system. Naive B cells, after encountering their cognate antigen, migrate to GCs, where they alternate cycles of affinity-based selection (in the GC's light zone) and proliferation and mutation (in the dark zone) (*Cyster and Allen, 2019*; *Victora and Nussenzweig, 2022*). Over time scales of several weeks (and many generations), this results in an increase in the average affinity of the population of B cells (*Berek and Milstein, 1987*; *Allen et al., 1988*). Understanding these processes that occur in the GC is thus of great importance

both to achieving a fundamental grasp of the immune system and to advancing goals such as making a vaccine for difficult-to-neutralize viruses (*Burton et al., 2012*).

The GC is a Darwinian evolutionary system designed to improve affinity, and as such, one might ask: What relationship between affinity and fitness does it use? In order for the GC reaction to improve affinity, this relationship must be increasing. That is, on average, B cells with higher affinity BCRs will reproduce more than those with low affinity BCRs, leading to an overall improvement in affinity. However, the details of the relationship between affinity and fitness are unknown. This is in part because the fitness of a B cell is an emergent property of an intact, functioning germinal center and thus cannot be measured using an in vitro assay.

There have been a variety of suggestions for the form of the affinity–fitness relationship (*Batista and Neuberger, 1998*; *Kuraoka et al., 2016*; *Murugan et al., 2018*). Resolving these possibilities in practice, however, requires care, even in the ideal situation of single-cell sequencing of individual germinal centers (*Tas et al., 2016*; *DeWitt et al., 2025*). One cannot simply examine by hand trees built from observed sequences and gain insight. Indeed, processes such as the 'push of the past' and the 'pull of the present' (*Nee et al., 1994*; *Budd and Mann, 2018*) can mislead interpretation of germinal center sequence data. Instead, an approach is needed that can model extinct lineages that do not appear in the reconstructed tree (*DeWitt et al., 2025*).

We will approach the affinity–fitness relationship via a birth-death model in which the birth rate is a function of affinity; we call this function the 'affinity–fitness response function'. We assume a family of such functions, determined by some set of parameters, such that a choice of parameters gives a specific response function. We will use data from the 'replay' experiment (*DeWitt et al., 2025*) to fit this function. This experiment used mice whose GCs are seeded entirely by B cells with a single, fixed naive antibody. They were immunized with this naive antibody's cognate antigen, and the subsequent GC reactions were observed in two ways: by extracting and sequencing individual GCs near a single timepoint, and by pooling multiple GCs at each of a wide range of timepoints for single-cell sequencing. In addition, a deep mutational scan (DMS) on the naive antibody's sequence was performed for binding to the immunogen. This DMS experiment enables us to make predictions about the affinity of an arbitrary sequence. These affinities can then be input into the affinity–fitness response function; one can compare the observed phylogenetic trees to predictions made by the corresponding birth-death process to learn about the parameters of the response function.

Ideally, one would be able to fit the parameters of the affinity–fitness response function via maximum likelihood or Bayesian inference. However, this is difficult. In general, calculating likelihoods for birth-death models is a challenge, requiring solutions of ordinary differential equations (ODEs) that take the probability of unsampled lineages into account. *Bakis et al., 2025* have employed this approach, using the multitype birth-death (MTBD) model of *Kühnert et al., 2016*; *Barido-Sottani et al., 2020* but adapting it to the case of many trees evaluated in parallel. The types in the MTBD are discretized bins of affinity, and mutation moves lineages between bins.

This approach has the substantial advantage of being rigorously tractable using numerical integration of ODEs and Markov chain Monte Carlo (MCMC); however, the inferential model is misspecified in two ways. First, it requires Markov evolution of the types in the MTBD model. In our case, the type is determined by the sequence, and so type evolution is determined by sequence evolution. Because the relationship between sequence and affinity type is non-trivial, and the sequence-based mutation process is complex, evolution of the types is not Markovian. Second, the MTBD model cannot accommodate population size constraints such as those present in real germinal centers. GCs typically reach carrying capacity around 10 days, well before our sampling for these experiments at 15 or 20 days (*Schwickert et al., 2007*; *Mesin et al., 2020*). Models with such constraints, in fact, generally have intractable likelihoods because the evolution of each cell depends on the entire population of cells.

Likelihood-free inference is an alternative approach to fitting complex statistical models. Classical likelihood-free inference uses Approximate Bayesian Computation with summary statistics (*Beaumont et al., 2002*). More recently, techniques using deep neural networks have emerged, especially in population genetics (*Sheehan and Song, 2016*; *Schrider and Kern, 2018*). Recent work has extended this to the inference of phylogenetic model parameters (*Voznica et al., 2022*; *Lajaaiti et al., 2023*; *Landis and Thompson, 2025*) including state-dependent diversification models (*Lambert et al., 2023*; *Thompson et al., 2024*). Can we use these recent methods to learn the parameters of the affinity–fitness function in the germinal center?

In this paper, we learn the relationship between affinity and fitness via simulation-based inference using cell-based forward simulation, neural network inference, and summary statistics matching (*Figure 1*). We begin from the compact bijective ladderized vector (CBLV) tree encoding of *Voznica et al., 2022*, adding affinities for each node from ancestral sequence reconstruction and the sequence-affinity mapping. Our forward model contains too many parameters to be inferred using neural networks alone, so we complement the neural network inference with summary statistic matching for some parameters. We infer response functions on 119 trees from *DeWitt et al., 2025* and find them to be generally similar across GCs, and roughly tripling from affinity 0 (naive) to 1, and then tripling again from 1 to 2. This means that a cell that mutated to affinity 1 in the early stages of the GC would replicate roughly three times as fast as the cells around it.

## Methods
### Overview of the replay experiment and data
In order to quantify the mechanics of GC selection, we use data from an experimental mouse system that we nickname the 'replay' experiment (*DeWitt et al., 2025*) in which all naive B cells seeding GCs are identical, carrying the same pre-rearranged IG genes. Thus, the starting sequences for affinity maturation in these GCs have identical affinity and specificity for their cognate antigen. This antigen is chicken IgY, which has been characterized previously (*Tas et al., 2016*; *Jacobsen et al., 2018*). The naive sequence chosen (called 'clone 2.1') was the naive precursor to the dominant sequence in a GC with a very large clonal burst 10 days after immunization (*Tas et al., 2016*), suggesting it is a good starting point for affinity maturation. The engineered mice carry the unmutated versions of the Igh and Igk genes rearranged by clone 2.1 in their respective loci. They were immunized with chicken IgY, and the resulting GC reactions were observed in two ways, resulting in two separate data sets. For the first, which we refer to as the 'extracted GC data', 119 individual GCs were extracted and sequenced from 12 such mice at either 15 days (52 GCs) or 20 days (67 GCs) after immunization. In the second, called 'bulk data', multiple GCs from the whole lymph nodes of several mice were pooled together at each of seven timepoints (from 5 to 70 days after immunization), and then analyzed with droplet-based sequencing. In this paper, we only use the extracted GC data; however, we compare our results to others derived from the bulk data using separate inference methods.

We infer trees with IQ-TREE (*Minh et al., 2020*) version 1.6.12 using all observed BCR sequences from each GC with the known naive sequence as outgroup, and ancestral sequence reconstruction by empirical Bayes. We use the branch lengths from IQ-TREE directly, without attempting to infer calendar times for the internal nodes.

These BCR trees are annotated with antibody affinities at all nodes as follows. First, we perform ancestral sequence reconstruction for every node of the phylogenetic tree. Because these trees have relatively few mutations, this can be done with low error. We then take affinity values for each single-mutation variant from a DMS experiment (*DeWitt et al., 2025*) that used methods similar to *Adams et al., 2016*. For sequences with more than one mutation, we simply add the effect of each individual mutation (i.e. we assume no epistasis), which was shown to be a reasonable approximation in *DeWitt et al., 2025*.

Affinity is defined relative to the naive dissociation constant ($K_D^N \simeq 40nM = 4x10^{-8}M$ *Tas et al., 2016*):

$$x = -\log_{10} K_D/K_D^N \tag{1}$$

such that, for example, an affinity of +2 means a $K_D$ 0.01 times the naive value. Recall that lower $K_D$ means stronger binding, which corresponds to larger $x$. Affinity is 0 for the initial unmutated sequence and ranges from −12.2 to 3.5 in observed sequences, with a mean (median) of −0.3 (0.3).

The 119 trees have between 26 and 95 leaves, with a mean (median) of 74 (78) leaves. Observed sequences have between 0 and 19 nucleotide mutations, with a mean (median) of 6.3 (6.0), or around 1%. It is important to note that this is a much lower level than typical BCR repertoires, which average roughly 5–10% nucleotide SHM.

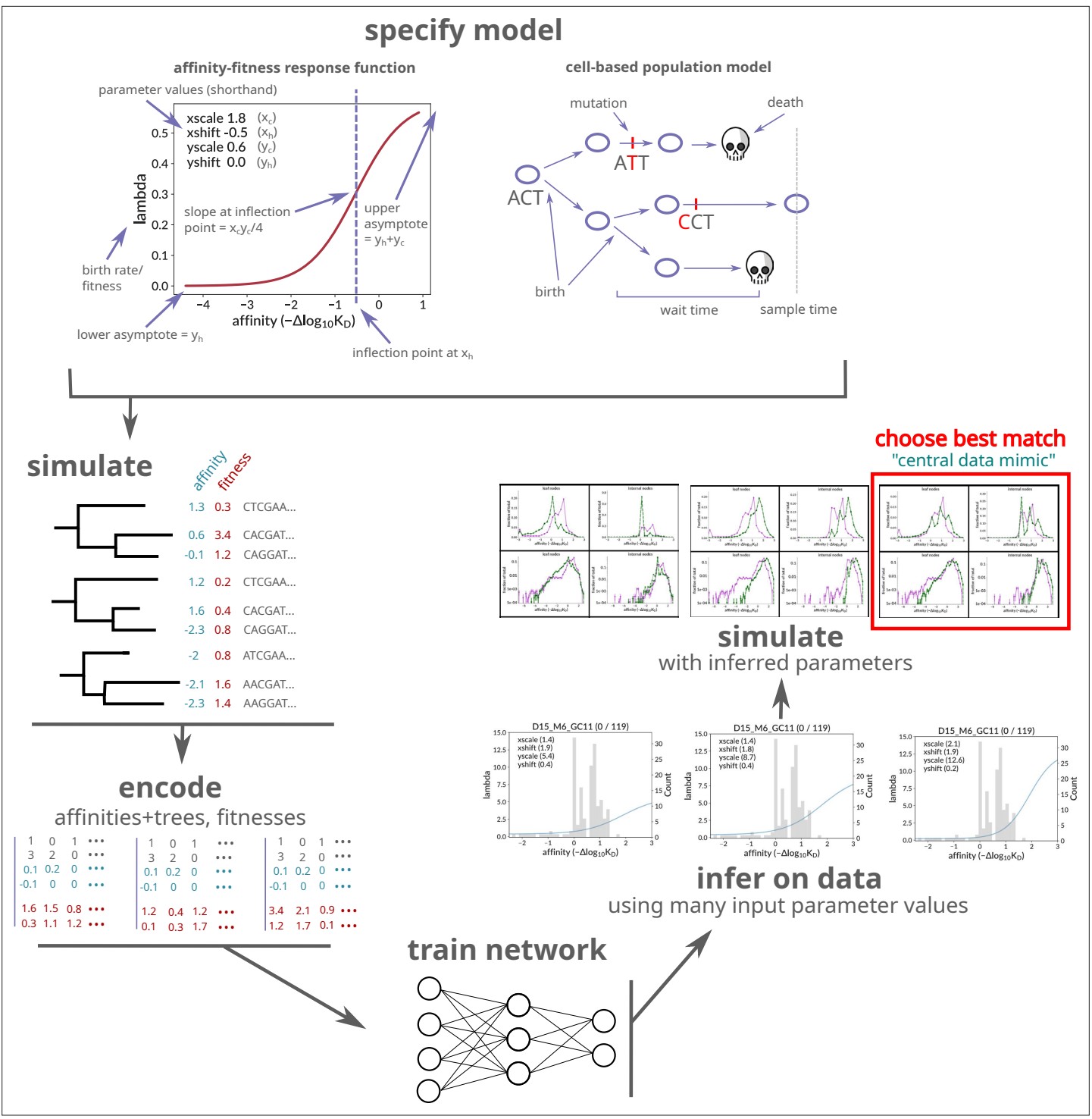

**Figure 1.** Overview of the workflow for this paper. After specifying a birth-death model with sigmoid affinity–fitness response function, we simulate many trees and their sequences at each node, with parameters roughly consistent with our real data samples. The simulation is cell-based and implements a carrying-capacity population size limit. The results of the simulation are then encoded and used to train a neural network that infers the sigmoid response parameters on real data. In addition to encoded trees, the network also takes as input the assumed values of several non-sigmoid parameters (carrying capacity, initial population size, and death rate). Inference on real data is performed many times, with many different combinations of non-sigmoid parameter values, and additional 'data mimic' simulation is generated using each of the resulting inferred parameter value combinations. The summary statistics of each data mimic sample are compared to real data, with the best match selected as the 'central data mimic' sample with final parameter values for both sigmoid (inferred with the neural network) and non-sigmoid (inferred by matching summary statistics)

*Figure 1 continued on next page*

parameters. Plots from elsewhere in the paper are rendered in schematic form: those in 'infer on data' refer to *Figure 4—figure supplement 1*, and those in 'simulate with inferred parameters' to *Figure 5*.

The online version of this article includes the following figure supplement(s) for figure 1:

**Figure supplement 1.** Simulation response function example with sampled affinity values.

**Figure supplement 2.** Diagram of curve difference loss function calculation.

**Figure supplement 3.** Example of approximate sigmoid parameter degeneracy.

## Model

In order to simulate sequences and trees that we use to train the neural network, we employ a birth-death model with a population size constraint. Our model is designed to mimic the replay experiment, with a parameterized affinity–fitness response function mapping each cell's affinity to its fitness. The simulation is initialized with a single root node with the naive sequence from the replay experiment. This initial node then immediately undergoes repeated binary expansion with neither mutation nor time passage until we have the desired number of initial naive sequences.

Each subsequent step in the simulation process consists of choosing one of the three possible events (birth, death, mutation) as follows. We loop over all event types and all nodes, calculating a waiting time for each such node-event type combination. These waiting times are calculated from an event type-specific response function, which relates each node's affinity to the likelihood, or rate, of that event type. For instance, the birth response function relates the node's affinity to the rate at which it divides (i.e., at which birth occurs).

The waiting time for each node-event type combination is sampled from the reciprocal of the response function (since response functions are rates, their reciprocal has units of time). For mutation events, the response function is calculated using the SHM model mutabilities summed over each node's sequence. Death events use a two-category response function, with separate (constant) rates for functional and non-functional sequences to reflect rapid purging of nonfunctional cells in the dark zone. The event type-node combination with the smallest waiting time is then selected to occur, and the corresponding cell is removed from the population. If a birth event is selected, two identical children are added to the population. For a death event, no cells are added. For a mutation, the location of the substitution and the new base are chosen according to probabilities from 5-mer sequence contexts in the node's sequence using the model of *Cui et al., 2016*; a single new cell is created to replace the old.

The process is stopped after a fixed period of time, and a sample of a given size is taken uniformly. If the number of living cells falls to zero, or if there are fewer than 10 living cells at sampling time, the tree is discarded and the simulation is retried.

Inspired by *Dessalles et al., 2018*, we enforce a carrying capacity on the GC by logistically modulating the birth rate such that the process is critical (average birth and death rates over living cells are equal) when the population is at carrying capacity. We modulate each cell's birth rate by applying a multiplicative factor $m$:

$$m = \left( \frac{\sum_i \mu_i}{\sum_i \lambda_i} \right)^{N/N_0} \tag{2}$$

which, for carrying capacity $N_0$, sums death rates $\mu_i$ and birth rates $\lambda_i$ over the $N$ living cells. For initially small populations, the exponent is near zero and no modulation occurs. As the population increases, however, the exponent goes to one. Thus, as evolution increases affinities and, consequently, birth rates (in the denominator), $m$ becomes small, decreasing each $\lambda_i$. Thus, initial exponential growth (a supercritical process) gradually slows as it approaches criticality at the specified capacity. If the population fluctuates above carrying capacity, the exponent $N/N_0 > 1$ causes $m$ to decrease rapidly, which in turn decreases the birth rates in its denominator.

We specify the birth response function (relating a node's affinity to its birth rate) as a sigmoid on affinity $x$ (*Figure 1—figure supplement 1*):

$$\lambda(x) = \frac{y_c}{1 + e^{-x_c(x - x_h)}} + y_h. \tag{3}$$

**Table 1.** Simulation parameter values for the sample used for training (left column, in which each GC has different parameters), and the 'central data mimic' sample used to evaluate final results (right column, in which all GCs have the same, data-inferred parameters).

Square brackets indicate a range, from which values are chosen either as detailed in the text (for sigmoid parameters) or uniformly at random. The mutability multiplier is an empirical factor that modulates the intensity of mutation to more closely match the speed of evolution to that observed in data.

| | Training | Central data mimic |
|---|---|---|
| Birth response function | Sigmoid | Sigmoid |
| xscale ($x_c$) | [0.01, 2] | 1.6 |
| xshift ($x_h$) | [–0.5, 3] | 2.0 |
| yscale ($y_c$) | [0.5, 35] | 18.2 |
| yshift ($y_h$) | [0, 0.6] | 0.4 |
| Carrying capacity | [500, 2000] | 500 |
| Capacity method | birth | birth |
| Time to sampling | [10, 35] | 20 |
| # sampled cells/GC | [50, 130] | [60, 95] |
| # GCs | 50,000 | 120 |
| Naive birth rate | [0.1, 15] | 1.2 |
| Death rate (functional) | [0.05, 0.5] | 0.2 |
| Death rate (stops) | 10 | 10 |
| Initial population | [8-128] | 128 |
| Mutability multiplier | 0.68 | 0.5 |

The sigmoid's upper asymptote ($y_h + y_c$) represents a hypothesized fitness ceiling above which further affinity increases do not improve fitness. Its lower asymptote ($y_h$) gives the fitness of very low-affinity nodes (due, for instance, to tonic signaling), which represent cells with either nonfunctional or severely compromised receptors. The $x$ position of the transition between these asymptotes is given by $x_h$, while $x_c$ determines the transition's steepness. When plotting, we use the following more descriptive names:

- $x_h \rightarrow$ xshift
- $x_c \rightarrow$ xscale
- $y_c \rightarrow$ yscale
- $y_h \rightarrow$ yshift

To construct our training sample, we want to sample $x_c$, $x_h$ $y_c$, and $y_h$ values from throughout biologically plausible ranges. One cannot, however, choose the value of each parameter independently without yielding unrealistic initial birth rates. Besides poorly describing real GCs, such rates would result in many failed simulations. Along with bounds on each of these parameters, we thus also incorporate bounds on the naive (zero-affinity) birth rate, namely

$$\lambda_0 = \frac{y_c}{1 + e^{x_c x_h}}, \tag{4}$$

with lower and upper bounds $\lambda_0^l$ and $\lambda_0^u$. Note that in deriving *Equation 4* we have neglected $y_h$, and instead choose $y_h$ within its own bounds, independently of the other parameters. This amounts to the approximation that very low-affinity nodes have much lower fitness than those with high affinity ($y_h << y_c$). We then choose each remaining parameter in an arbitrary, but consistent, order with the following procedure. We first choose a value for $x_c$ from within its bounds $[x_c^l, x_c^u]$ (*Table 1*), and substitute that value into the $\lambda_0$ constraints. This gives us an additional constraint on the next parameter, $x_h$, which also has its own bounds $x_h^l$ and $x_h^u$:

$$\frac{\log\left(\frac{y_c^l}{\lambda_0^u} - 1\right)}{x_c} < x_h < \frac{\log\left(\frac{y_c^u}{\lambda_0^l} - 1\right)}{x_c}. \tag{5}$$

Analogous additional constraints are then also incorporated when subsequently choosing the last parameter $y_c$, with its own bounds $y_c^l$ and $y_c^u$:

$$\lambda_0^l(1 + e^{x_c x_h}) < y_c < \lambda_0^u(1 + e^{x_c x_h}). \tag{6}$$

Values for all simulation parameters are listed in *Table 1*.

## Tree encoding

We encode each tree with an approach similar to *Lambert et al., 2023* and *Thompson et al., 2024*, most closely following the compact bijective ladderized vector (CBLV) approach from *Voznica et al., 2022*. The CBLV method first ladderizes the tree by rotating each subtree such that, roughly speaking, longer branches end up toward the left. This does not modify the tree, but rather allows iteration over nodes in a defined, repeatable way, called inorder iteration. To generate the matrix, we traverse the ladderized tree inorder, calculating a distance to associate with each node. For internal nodes, this is the distance to root, whereas for leaf nodes, it is the distance to the most-recently-visited internal node (*Voznica et al., 2022*, *Figure 2*). Distances corresponding to leaf nodes are arranged in the first row of the matrix, while those from internal nodes form the second row.

We further enrich the basic CBLV encoding with information on the state of the evolving entity. In contrast to *Lambert et al., 2023* and *Thompson et al., 2024*, however, who labeled only tips with discrete types, we label all nodes with a continuous variable. The variable we are interested in is affinity, which we incorporate into the encoded matrix as two additional rows. Since the original matrix's two rows represent leaf and internal node heights, each entry in our additional two rows represents the corresponding node's affinity value (*Figure 1*).

In addition to repeatability, we want the node ordering to guarantee a unique location for each node in the resulting matrix. In the original method (*Voznica et al., 2022*), however, this correspondence (and thus the encoding) is only unique if all nodes have unique times. For generality, we would like to allow the use of ultrametric trees, with all cells sampled at the same time. This has the potential

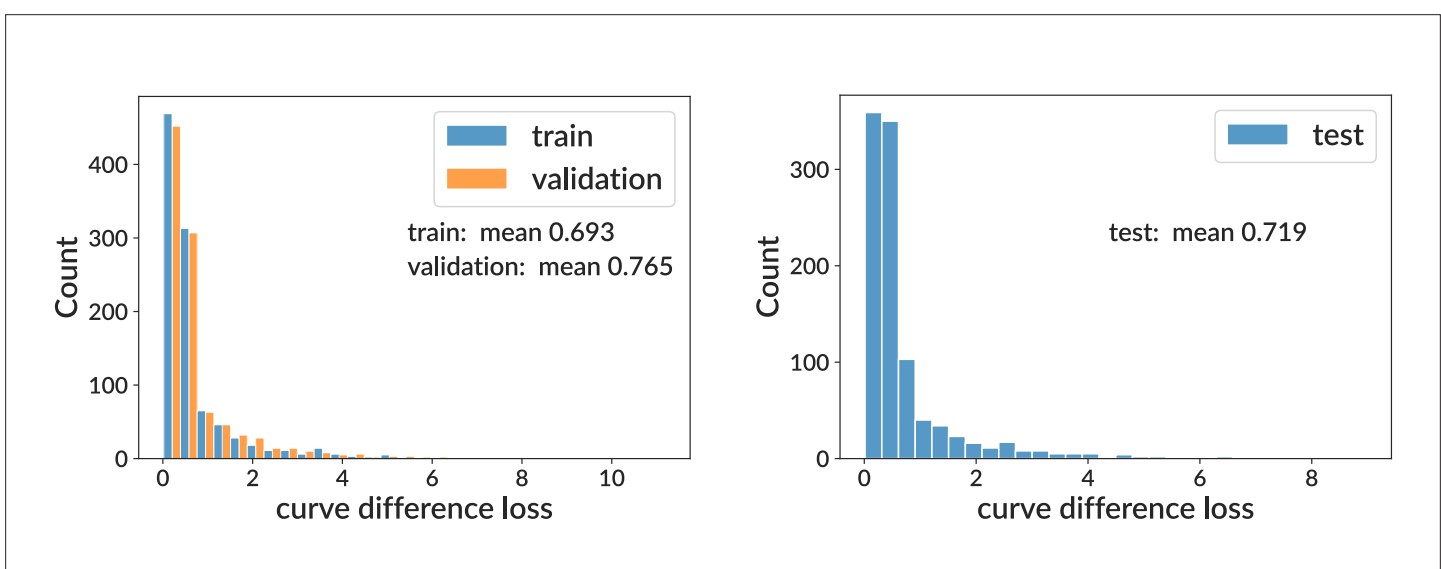

**Figure 2.** Training and testing results for the sigmoid model on simulation. We show curve difference loss distributions on several subsets of the training sample (where each GC has different parameter values): training and validation (left) and testing (right). For computational efficiency when plotting, the curve difference distributions display only the first 1000 values. See *Figure 3* for per-bin model.

The online version of this article includes the following figure supplement(s) for figure 2:

**Figure supplement 1.** Example true and inferred response functions on the training sample.

**Table 2.** Neural network architecture.

| Order | Type | Filters | Kernel size | Pool size | Stride length | N units | N params |
|---|---|---|---|---|---|---|---|
| 1 | 1D convolutional | 25 | 4 | | | | 426 |
| 2 | 1D convolutional | 25 | 4 | | | | 2525 |
| 3 | Max pooling | | | 2 | 2 | | |
| 4 | 1D convolutional | 40 | 4 | | | | 4040 |
| 5 | Global avg. pooling | | | | | | |
| 6 | Dense | | | | | 48 | 1968 |
| 7 | Dense | | | | | 32 | 1568 |
| 8 | Dense | | | | | 16 | 528 |
| 9 | Dense | | | | | 8 | 136 |
| 10 | Dense | | | | | 3 | 27 |

to confuse the neural network, since it destroys the one-to-one correspondence between trees and encodings. We thus add several tiebreakers to the ladderization scheme that establishes the order in which we iterate over nodes. After first sorting nodes by their time, we then sort any ties by the time of their immediate ancestral node, and then further sort any remaining ties by their affinity. This could, in principle, still result in ties. In practice, however, we have set our code to raise exceptions if ties are encountered, and this has not yet occurred.

As for real data, we infer trees on simulation with IQ-TREE (*Minh et al., 2020*). By taking only sampled sequences and passing them through the same inference method, we treat data and simulation as similarly as possible, which facilitates accurate neural network training and summary statistic comparison. An alternative would be to compare time trees inferred on data, for instance with BEAST (*Suchard et al., 2018*), to simulation truth trees. Although this is commonly done (*Voznica et al., 2022*), we reasoned that inference would be more accurate with simulation and data samples on the same footing. Furthermore, our implementation of initial population size creates large numbers of unmutated ancestral sequences near the root of the simulation truth tree, which would have no observable counterpart in data, where we can only infer the presence of common ancestor nodes.

As in *Voznica et al., 2022*, we also scale trees to mean unit depth before encoding in order to facilitate comparison across trees of different depths.

## Neural network architecture, implementation, and training

Our network is detailed in *Table 2*, but in brief, after input of encoded trees, we begin with several convolutional layers interspersed with pooling layers. There follows a series of dense layers with decreasing complexity until we arrive at the final number of outputs. These dense layers take as input, in addition to the output of previous layers, fixed values for several 'non-sigmoid' parameters (i.e., parameters not pertaining to the sigmoid birth response whose parameters we directly infer with the network). The input matrices are padded with zeros to a constant, uniform size (here set to 200), to allow the use of trees with different numbers of nodes. We use the exponential linear unit (ELU) activation function for all layers. Our main 'sigmoid' network predicts the four sigmoid parameters. However, as a cross-check, we also train a model that predicts the actual response function value in discrete bins of affinity, which we call the 'per-bin' model. The per-bin model introduces some level of

**Table 3.** Clip function bounds for neural network training.

| Parameter | Min value | Max value |
|---|---|---|
| xscale ($x_c$) | 0.001 | 3.5 |
| xshift ($x_h$) | −1.5 | 5 |
| yscale ($y_c$) | 0.1 | 65 |
| yshift ($y_h$) | 0 | 10 |

independence from the sigmoid model; however, because it is still trained on sigmoid simulation, it has only a limited ability to infer shapes that differ significantly from a sigmoid.

In making our training samples, we endeavor to cover the entire space of plausible parameter values, which includes choosing bounds on each of the three sigmoid parameters (*Table 1*). We find that leaving the inferred parameters entirely unconstrained tends to confuse the neural network, so we impose a 'clip function' with bounds on each parameter during network training. For decent performance, these clipping bounds must be somewhat wider than the simulation bounds; but, heuristically, not overly wide (*Table 3*).

We scale all input variables (branch lengths, affinities, and non-sigmoid parameters) individually to mean 0 and variance 1 over all trees. We experimented with similarly scaling output parameters, but this did not improve performance.

We use an Adam optimizer with a learning rate of 0.01, an exponential moving average (EMA) momentum of 0.99, and a batch size of 32, values which were arrived at by performing hyperparameter optimization to minimize loss values. We do not use dropout regularization, since in our tests, it resulted in worse performance than using a validation sample to measure overfitting. When training, we use 20% of each sample for testing and also reserve 10% of the remaining 80% as a validation sample. We train for 35 epochs, which for both sigmoid and per-bin networks was the point at which validation sample performance began to decrease. While during development we trained on a large number of simulation samples with different sizes and parameter ranges, the results presented in this paper center on a sample of 50,000 trees with the ranges specified in *Table 1*. We also used samples as large as 100,000 trees, and with substantially more complex networks, but neither change resulted in significant performance improvements. Also note that these optimization and testing steps were performed only on simulation; we ran data inference only once our methods were essentially finalized.

All code for this paper can be found at https://github.com/matsengrp/gcdyn (copy archived at *Ralph et al., 2026*) and all inputs and outputs, along with instructions for running, can be found at https://doi.org/10.5281/zenodo.15022130.

## Curve difference loss function

For a loss function, we use a scaled version of the $L^1$ distance between the true and inferred functions (*Figure 1—figure supplement 2*). This loss, which we refer to as a 'curve difference', is calculated by dividing the area between the curves by the area under the true curve within domain, that is, affinity bounds, $[-2.5, 3]$. This domain was chosen because below $-2.5$ most curves are very similar, while above 3 they diverge dramatically in a region in which we have very few measured affinity values. The per-bin model uses the same loss function, but with a coarser discretization (one value for each bin).

An alternate approach would have been to use mean squared error on the inferred parameters, namely the squared difference between inferred and true values over all predictions. In our case, however, this is a poor choice because we care about the sigmoid's shape, but not about the underlying sigmoid parameter values, which have little individual biological meaning. The curve loss and the mean squared loss on inferred parameters can be quite different: significant simultaneous changes to two parameters can counteract each other, resulting in very similar curve shapes (*Figure 1—figure supplement 3*).

## Non-sigmoid parameter inference using summary statistics

In addition to the four sigmoid parameters, which we infer directly, there are other parameters in *Table 1* about which we have incomplete information. The carrying capacity method and the choice of sigmoid for the response function represent fundamental model assumptions. We also fix the death rate for nonfunctional (stop) sequences, which would be very difficult to infer with the present experiment. For others, we know precise values from the replay experiment for each GC (time to sampling, # sampled cells/GC), but use a somewhat wider range for the sake of generalizability. The mutability multiplier is a heuristic factor used to match the SHM distributions to data. The naive birth rate is determined by the sigmoid parameters, but has its own range in order to facilitate efficient simulation.

For two of the three remaining parameters (carrying capacity and initial population), we can ostensibly choose values based on the replay experiment. These values carry significant uncertainty, however, partly due to inherent experimental uncertainty, but also because they may represent different biological quantities to those in simulation. For instance, an experimental measurement of

the number of B cells in a germinal center might appear to correspond closely to simulation carrying capacity. However, if germinal centers are not well mixed, such that competition occurs only among nearby cells, the 'effective' carrying capacity that each cell experiences could be much smaller.

Fortunately, in addition to the neural network inference of sigmoid parameters, we have another source of information that we can use to infer non-sigmoid parameters: summary statistic distributions. We can use the matching of these distributions to effectively fit values for these additional unknown parameters. We also include the final parameter, the functional death rate, in these non-sigmoid inferred parameters, although it is unconstrained by the replay experiment, and it is unclear whether it is uniquely identifiable.

We adopt a two-step procedure to infer sigmoid and non-sigmoid parameters, inspired by the concept of conditional generation. Recall that our network takes non-sigmoid parameters as input, i.e. it infers sigmoid values contingent on a set of non-sigmoid values. We thus first infer sigmoid parameters on data for many different values of the non-sigmoid parameters. We then take each resulting set of (inferred) sigmoid and (supplied) non-sigmoid values, generate a new simulation sample with each, and compare their summary statistics to data. These simulation samples, like data but unlike the training sample, consist of 119 GCs with identical parameters and are referred to as 'data mimic' samples. The sample whose summary statistics most closely match data corresponds to the best inferred values for both sigmoid and non-sigmoid parameters and is called the 'central data mimic' sample. While we would ideally include all non-sigmoid parameters in this two-step procedure, the combinatorics of scanning over so many different variables is prohibitive. We thus only use three non-sigmoid parameters: carrying capacity, initial population, and death rate. These were chosen because their values likely include substantial uncertainty, and because they exert significant control over summary statistic distributions. This was implemented as a three-dimensional scan over four carrying capacity values (500, 750, 1000, 2000), three initial population values (8, 32, 128), and four death rates (0.05, 0.1, 0.2, 0.4).

Since the summary statistic distributions are affected by all simulation parameters, the best solution would likely be to infer all uncertain parameters with the neural network. This, however, would require passing much more information into a vastly more complicated network (the current tree encoding, for instance, has no information on sequence abundance) that would have to be trained on simulations covering additional dimensions. We thus deemed this approach infeasible for the current paper.

## Central (medoid) curve prediction

While our network predicts each sigmoid parameter individually, it is generally only useful to view the resulting curve as a whole. This is because many different sets of sigmoid parameter values can give quite similar shapes, since a change in one parameter can to some extent be compensated by modifying others (*Figure 1—figure supplement 3*). In order to present the results of all GCs together, we thus must compare predictions using curves, rather than individual parameters. We want to choose both a central, representative curve and associated uncertainty bounds that, for each prediction, incorporate all four predicted parameters at once. We do this by calculating the medoid curve among predictions, with distance defined as our curve difference loss function. We thus calculate, for the predicted curve for each GC, the sum of squared differences to all other predicted curves in the affinity bounds [−2.5, 3]. We then choose the curve with the smallest such sum as the medoid curve.

## Results
### Deep learning effectively infers parameters from simulated data

After building and training our neural network, we evaluate its performance on subsets of the training sample. While this evaluation provides an important baseline and sanity check, it is important to note that the training sample differs dramatically from real data (and the 'data mimic' simulation sample that mimics real data). While real data consists of 119 GCs with identical parameters and thus response functions, we need the GCs in our training sample to span the space of all plausible parameter values. This means that while we must evaluate performance on individual GCs in the training and testing samples, in real data (and data mimic simulation) we combine results from 119 curves into a central (medoid) curve. Inference on the training sample will thus appear vastly noisier than on real data and data mimic simulation and also cannot be plotted with all true and inferred curves together.

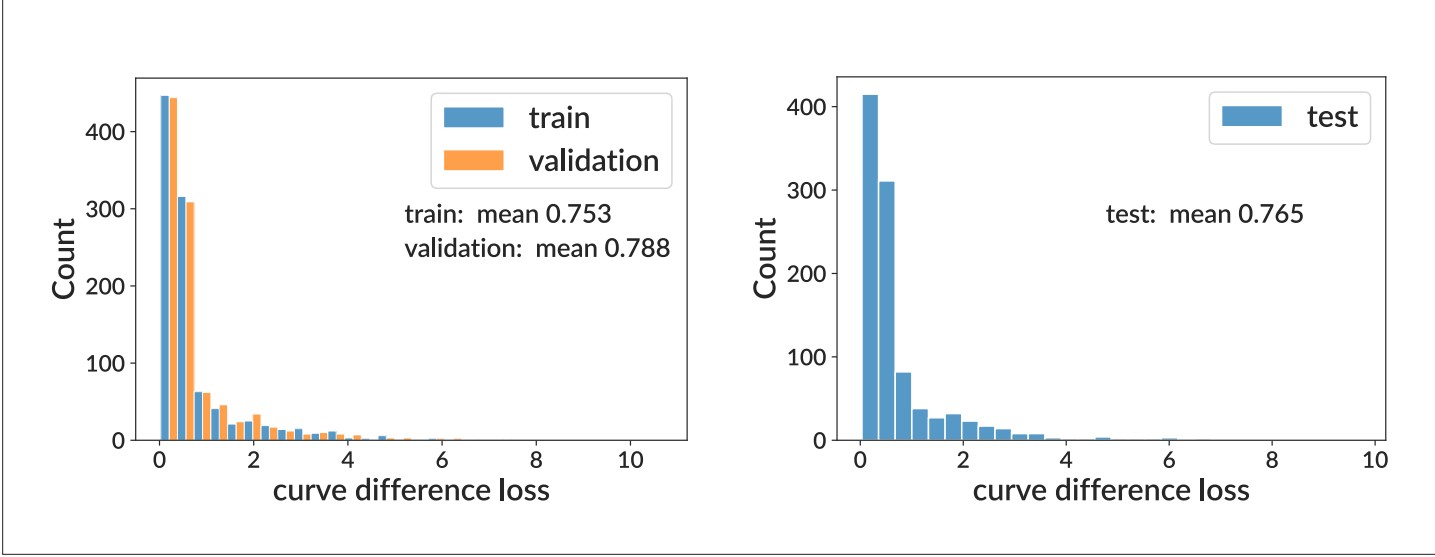

**Figure 3.** Training and testing results for the per-bin model on simulation. See *Figure 2* for details.

We show values of the curve difference loss function on the training sample (split into training, validation, and test subsets) for both sigmoid and per-bin models (*Figures 2 and 3*). This shows that for both models, the mean area difference on a single GC between true and inferred curves is around 70% of the full plot area. We find that the sigmoid and per-bin models have similar overall performance. We also show true and inferred response functions on four representative GCs with loss values spanning 0.17–2.18 (*Figure 2—figure supplement 1*).

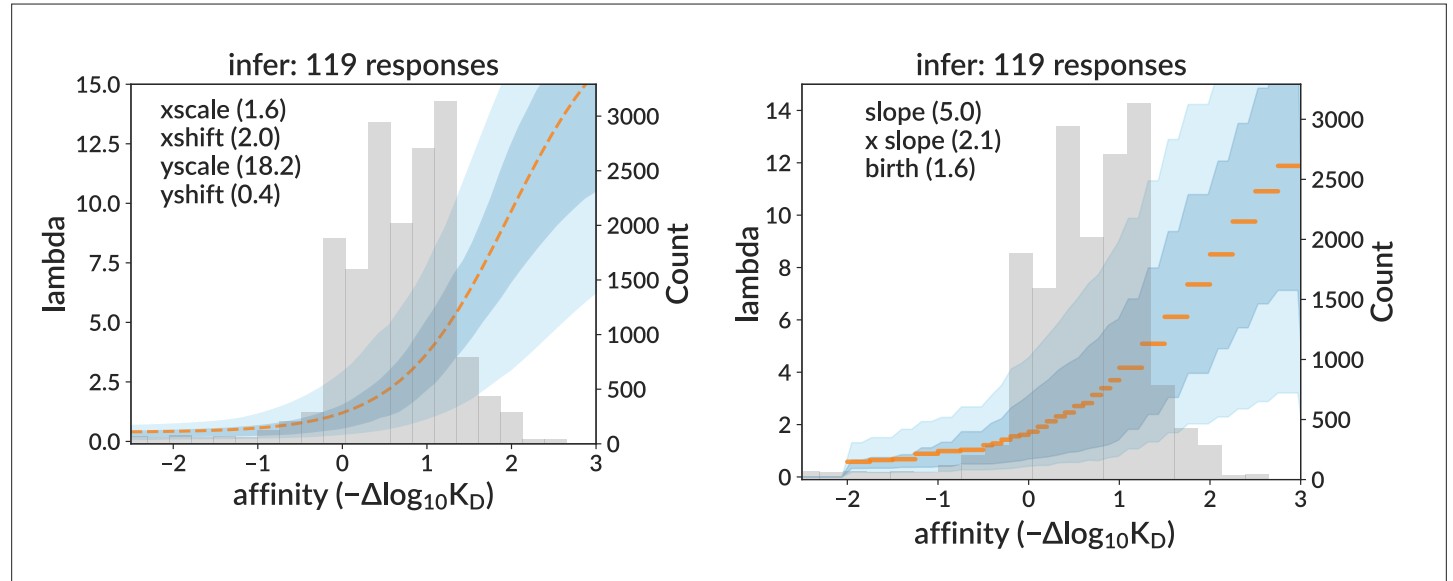

**Figure 4.** Inferred response functions on real data for sigmoid (left) and per-bin (right) models, corresponding to the non-sigmoid parameter values yielding simulation with the best-matching summary statistics. The medoid curve is shown in orange with 68% and 95% confidence intervals in blue, with observed affinity values in gray.

The online version of this article includes the following figure supplement(s) for figure 4:

**Figure supplement 1.** Example inferred sigmoid curves on data for four representative GCs.

**Figure supplement 2.** Example inferred per-bin response functions on data for four representative GCs.

## Inference on real data

We then apply our neural network to real data. We show the inferred sigmoid response curves on all 119 GCs for both the sigmoid and per-bin models (*Figure 4*), as well as several representative individual curve predictions (*Figure 4—figure supplements 1 and 2*). As detailed in the 'Methods', we run data inference many times, for many different values of three non-sigmoid parameters, generate simulation from each combination, and compare the resulting summary statistics to real data. In *Figure 4*, we display inference corresponding to the best such parameter values, that is, the values yielding simulation whose summary statistics best matched data: carrying capacity 500, initial population 128, and death rate 0.2. The summary statistics for the resulting central data mimic sample are shown in *Figure 5*, *Figure 5—figure supplement 1*.

Finally, we also show curve inference results on the central data mimic sample (*Figure 6*). As expected, the curve difference loss values are vastly smaller than for the single-GC results on the training sample (where each GC had a different curve). Here, the medoid sigmoid curve has a loss of 0.09 (i.e., the true and inferred curves differ by 9% of the plot's area), compared to the mean single-GC loss of 0.7 from the training sample. Also importantly, the variance of inferred curves (blue shading) is of similar magnitude to that in data.

## Simulation mimics GC dynamics

In addition to letting us infer non-sigmoid parameter values, the summary statistics distributions also establish how well our simulation mimics real BCR sequencing data. Since for the purposes of this paper we are interested in replay-style experiments, we seek to mimic these particular samples. The summary statistics for the central data mimic sample (*Figure 5*) match data reasonably closely. The training sample, on the other hand, matches much less well because it is designed to span all plausible unknown parameter values to ensure that the neural network encounters any parameter combination that it is likely to see in data (*Figure 5—figure supplement 2*). The less-likely parameter combinations, of course, tend to yield poorly matching summary statistics.

## Effective birth rates

The birth rate described by the response function is an absolute quantity $\lambda$ that, in the absence of carrying capacity constraint, would be simply inverted to get the waiting time for birth events. With capacity constraint, however, we apply a scaling factor $m$ (*Equation 2*) before inversion. It can thus be useful to scale by $m$ and subtract the cell-specific death rate $\mu_i$ to calculate an 'effective' birth rate $m\lambda_i - \mu_i$ describing the net fitness advantage of cell $i$ at a given moment in time. We now call the original $\lambda$ the 'intrinsic' birth rate to distinguish it from this effective rate. We show this effective birth rate on several GCs with central data mimic simulation parameters that were allowed to persist to 50 days (far beyond the extracted GC data at 15 and 20 days, *Figure 7*).

The most striking feature of this effective rate is its variability: rather than being a fixed property of the GC system, it describes the evolutionary microenvironment of a particular GC at a particular moment in time. Overall, the effective rate has a much smaller scale, generally staying between −0.2 and 1, compared to 0 to more than 15 for the intrinsic rate. The effective rate is also time-dependent, with $m$ compressing its range as time goes on, which also contrasts with the fixed intrinsic rate. This compression continues well after the simulation has reached steady state population (blue line in right column, *Figure 7*), as the mean affinity (red line in right column) continues to move relative to the intrinsic birth rate's upper and lower asymptotes (middle column). Biologically speaking, this compression means that there is less reward for gaining affinity later in the simulation than in the beginning. This feature is shared with the traveling wave model (*DeWitt et al., 2025* Figure S6D).

In order to compare more directly to *DeWitt et al., 2025*, we remade their Figure S6D, truncating to values at which affinities are actually observed in the bulk data, and using only three of the seven timepoints (11, 20, and 70, *Figure 8*, left). We then simulated 25 GCs with central data mimic parameters out to 70 days. For each such GC, we found the timepoint with mean affinity over living cells closest to each of three specific 'target' affinity values (0.1, 1.0, 2.0) corresponding to the mean affinity of the bulk data at timepoints 11, 20, and 70. We then plot the effective birth rates of all living cells vs relative affinity (subtracting mean affinity) at the resulting GC-specific timepoints for all 25 GCs together (*Figure 8*, right). Note that because each GC evolves at very different and time-dependent rates, we could not simply use the timepoints from the bulk data, since each GC slice from

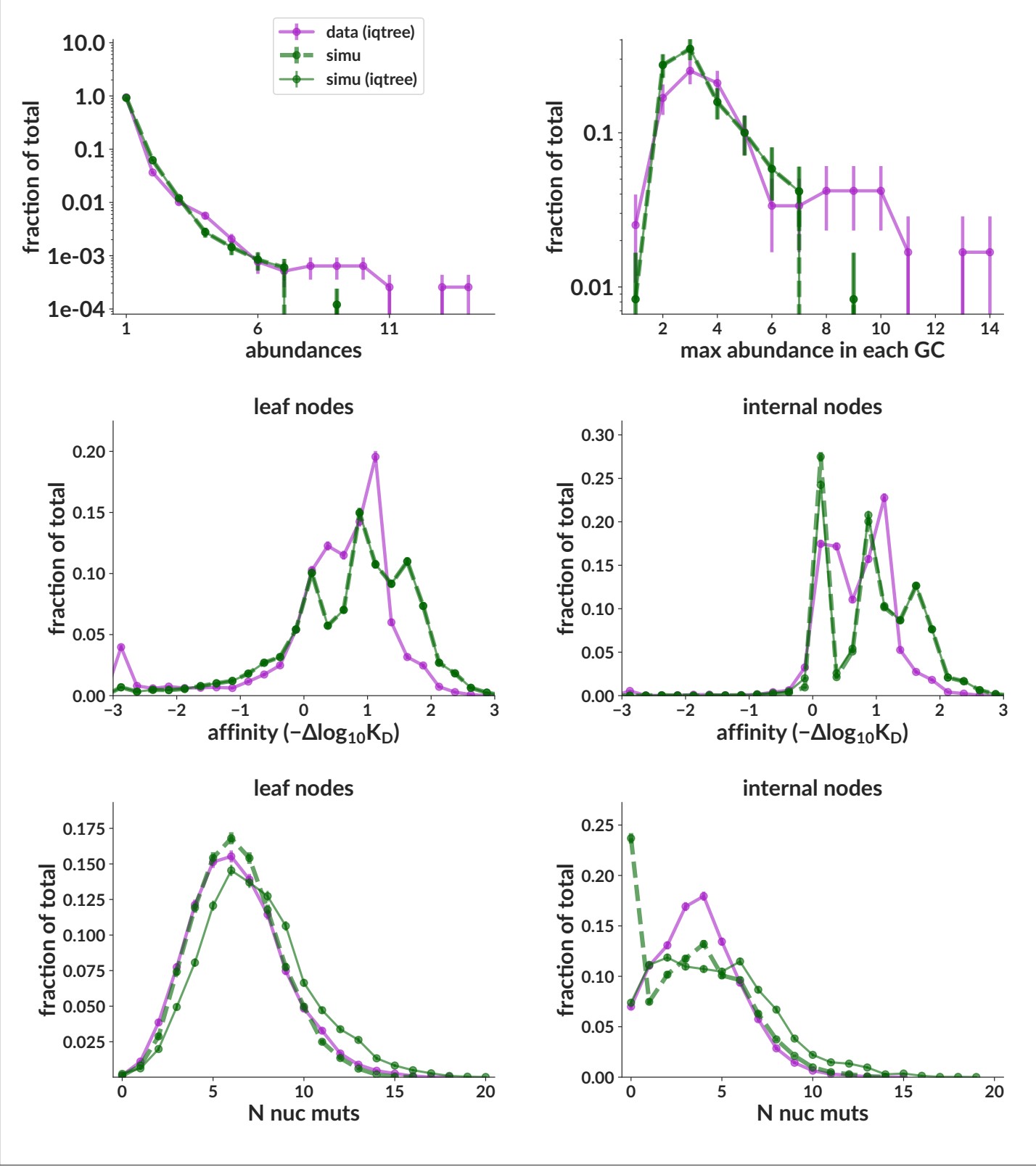

**Figure 5.** Summary statistics on data vs simulation for the central data mimic simulation sample that most closely mimics inferred data parameters. Simulation truth (dashed green) is unobservable and shown only for completeness; the important comparison is between purple and green solid lines, where both data and simulation have been run through IQ-TREE.

*Figure 5 continued on next page*

*Figure 5 continued*

The online version of this article includes the following figure supplement(s) for figure 5:

**Figure supplement 1.** Additional summary statistics distributions for the same central data mimic sample as the main figure.

**Figure supplement 2.** Summary statistic distributions for the simulation sample used for training.

our simulation would then have very different mean affinity. The mean over GCs of these GC-specific chosen times is 10.9, 24.5, 44.4 (compared to the original bulk data timepoints 11, 20, 70). It is important to note that while the first two target affinities (0.1 and 1.0) are within the affinity ranges encountered in the extracted GC data, the third value (2.0) is far beyond them, and thus represents extrapolation to an affinity regime informed more by our underlying model than by the real data on which we fit it.

Despite representing fundamentally different quantities, at positive relative affinities, the two estimates are relatively similar. At negative affinities, however, they diverge sharply, with the net growth rate from *DeWitt et al., 2025* falling without limit while our estimate of $m\lambda - \mu$ quickly plateaus. We defer a more extensive explanation of these features to the Discussion (see reference to *Figure 8*). In order to make the left-hand plot in *Figure 8*, we used the Python notebook from *DeWitt et al., 2025* here.

## Discussion

The germinal center reaction is a complex evolutionary process during which B cell populations improve their affinity. Little is known, however, about the specific form of the underlying relationship between affinity and fitness. Because a B cell's fitness depends on the context of other cells with which it interacts, this relationship cannot be measured directly in a lab. Its form must instead be inferred based on observations of intact, evolving germinal centers.

We model this biology using a birth-death-mutation process with a carrying capacity constraint. While birth-death models are common, the necessity of a capacity constraint makes likelihood evaluation intractable using existing methods (although recent theoretical work may provide a foundation for solving this in a 'mean-field' sense as described in *DeWitt et al., 2024*). We thus pursue a likelihood-free approach, training a deep neural network on simulation designed to emulate the replay experiment data. This likelihood-free approach also allows us to use a model of mutation on sequences that need not be Markov on affinity 'types'.

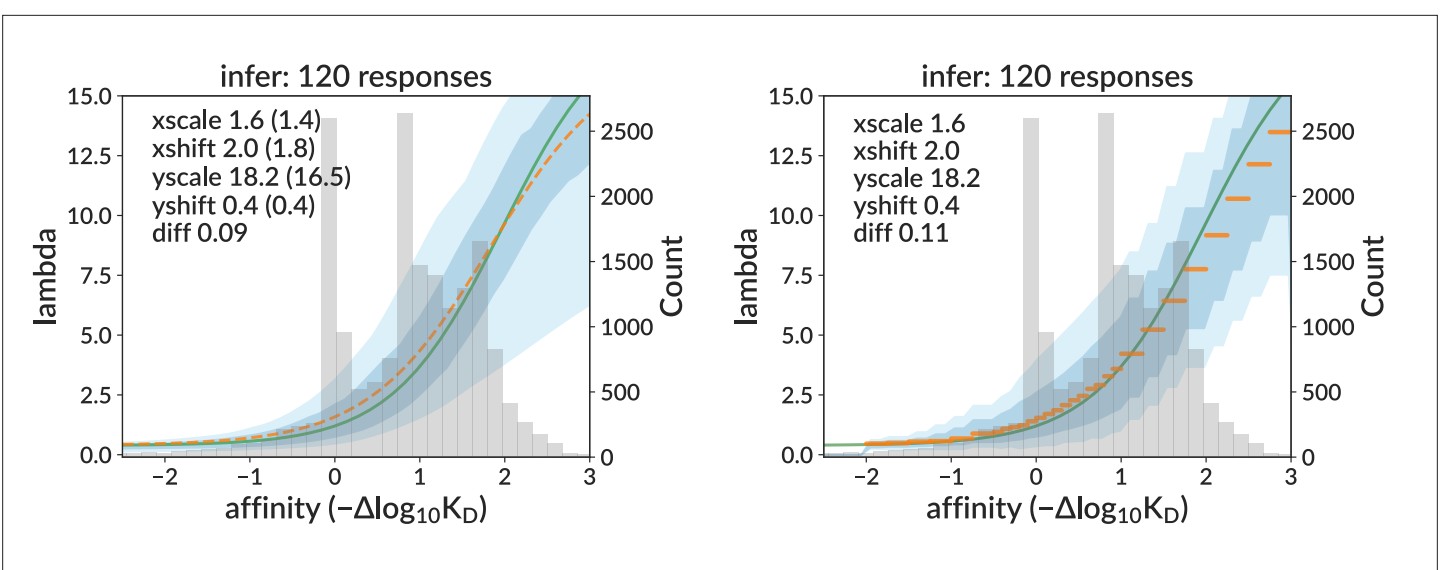

**Figure 6.** Inferred sigmoid curves for the central data mimic data-like simulation sample. The medoid curve is shown in orange, and the true curve is shown in green. This sample consists of 120 GCs all simulated with the same sigmoid parameters (from the central data prediction) and non-sigmoid parameters (from the best-matched summary statistics).

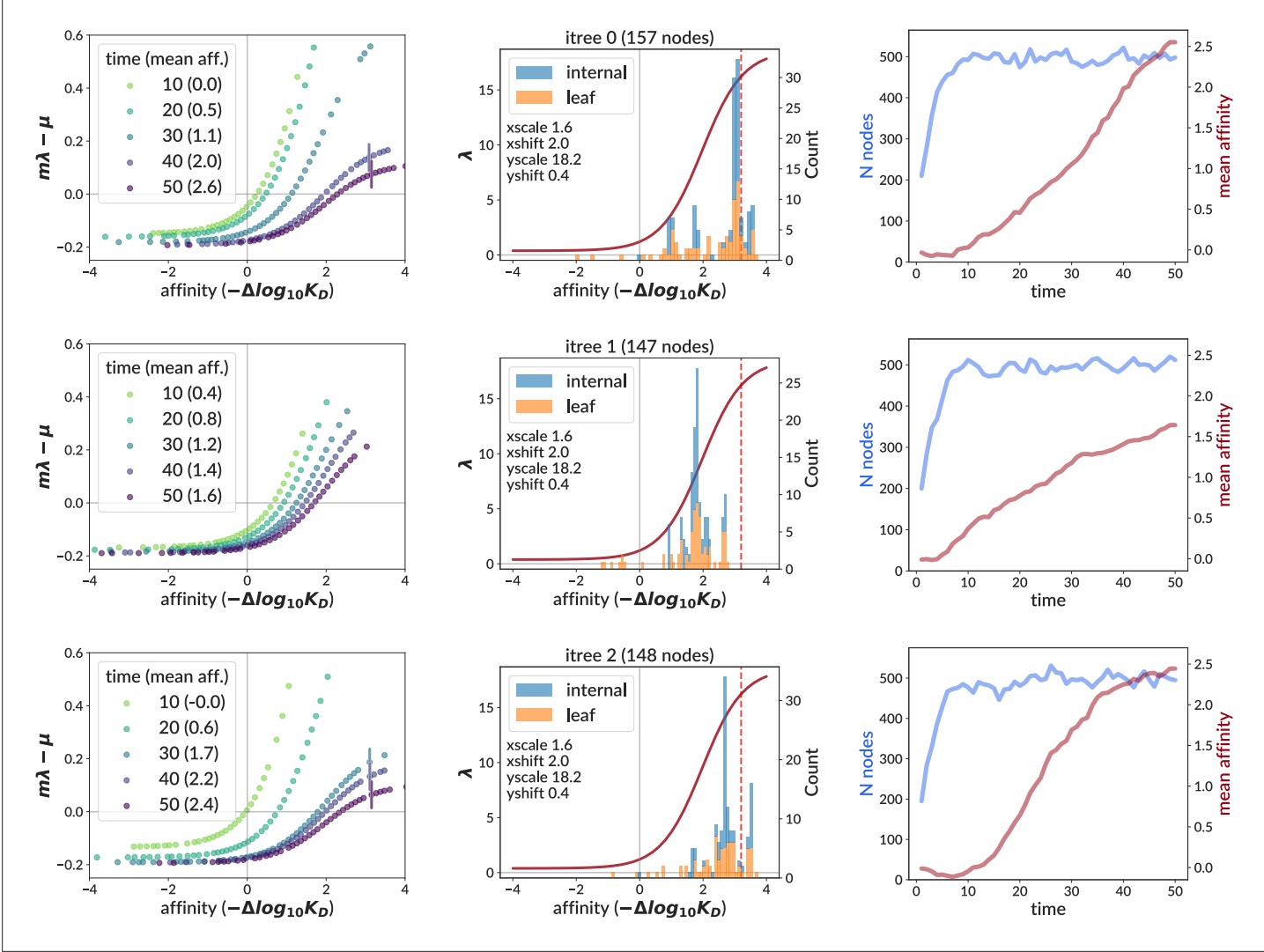

**Figure 7.** Effective birth rates (left column) calculated for three GCs simulated with the central data mimic parameters. The effective birth rate for cell $i$ is defined as $m\lambda_i - \mu_i$ for the carrying capacity modulating factor $m$, intrinsic birth rate $\lambda_i$, and death rate $\mu_i$. It is shown at several different time values for all living cells, except that for plotting clarity, cells closer to each other than 0.1 in affinity are not shown. Note that we extend time here to 50 days in order to aid comparison to the bulk data used by the traveling wave model (**DeWitt et al., 2025**, Figure S6D), which assumes a steady state. (Our extracted GC data is sampled at 15 and 20 days.) Vertical lines (left and middle columns) mark the point at which the slope has dropped to 1/2 its maximum value (if absent, the slope never falls below this threshold). We also show the intrinsic birth rate $\lambda$ (middle column, solid red curve), and include histograms of sampled affinities at the final timepoint. The right column shows time traces of the number of living cells (i.e. nodes, in blue) and the mean affinity of all living cells (red).

We used this neural network to infer the affinity–fitness response function. Because the model includes several parameters that are not well constrained by experiment, but whose prediction with the neural network would be infeasibly complicated, we adopted a two-step inference procedure. We first infer the sigmoid shape at many different potential non-sigmoid parameter values, then 'fit' the non-sigmoid parameters using summary statistics. In testing on simulation, the network showed the ability to reliably infer the sigmoid's shape on samples similar to the extracted GC data. When applied to that data, it inferred a consistent shape across trees that was in line with expectations from simulation, while also giving summary statistics well-matched to the data. This shape indicates that intrinsic fitness (defined as the rate of binary cell division before applying capacity constraint) roughly triples from affinity 0 (naive) to 1, and then triples again from 1 to 2. We did not observe any significant decrease in the response function slope (what we would call a ceiling on fitness) within the time range (20 days) of the GC extraction experiment, although we did see a decrease at affinities around 3.5,

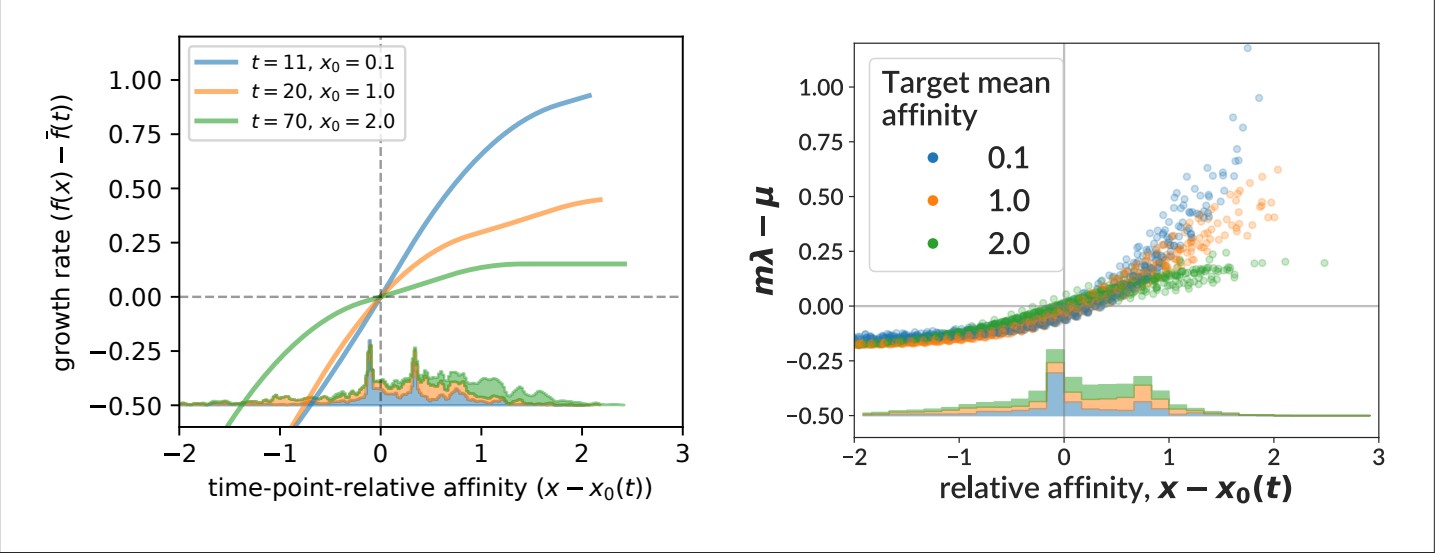

**Figure 8.** The net growth rate from **DeWitt et al., 2025** Figure S6D (left) compared to our effective birth rate (right) for three different 'target' mean affinity values. We selected three well-spaced timepoints from Figure S6D, and matched the corresponding mean affinity values from the bulk data to particular time slices in 25 GCs simulated with our central data mimic parameters (inferred on the extracted GC data) out to time 70. The net growth rate describes the net population growth at affinity $x$ and time $t$ in a traveling wave fitness model (see text). In our model, the effective birth rate is defined and plotted as in **Figure 7**. We also show stacked histograms of observed affinity values from the bulk data (left) and the 25 simulated GCs (*not* from the extracted GC data) (right). The measured affinity values from the extracted GC data (which extends only to time 20) are shown in **Figure 4**.

reached by day 50 in the data mimic simulation (**Figure 7**). This latter result should be treated with caution, however: this ceiling derives from an affinity region (see **Figure 4**) without observed affinities and is thus largely reflective of our assumed sigmoid shape, rather than any direct information from data. We also note that the extracted GC data has very low levels of mutation compared to typical BCR repertoires, so a ceiling may manifest only at higher affinity values than we observe.

## Prior work

### Measurements relevant to the affinity–fitness relationship

A variety of prior biological measurements help guide our expectations for the shape of the response function. The function's slope, for instance, controls the speed of evolution in the GC, with a steeper slope creating stronger clonal bursts and faster selective sweeps. We are not, however, aware of any work numerically quantifying the shape of this important curve other than our recent manuscript (**DeWitt et al., 2025**). The existence and potential location of a ceiling on fitness or affinity, on the other hand, has been the subject of much work. While affinity and fitness ceilings are separate concepts, they are closely related. An affinity ceiling is a limit to affinity for a given antigen: there are no mutations that can improve affinity beyond this level. This would result in a truncated response function, undefined beyond the affinity ceiling. A fitness ceiling, on the other hand, is an upper asymptote on the response function. Such a ceiling would result in a limit on affinity for a germinal center reaction, since once cells are well into the upper asymptote of fitness, they are no longer subject to selective pressure.

One study (**Batista and Neuberger, 1998**) measured how the B cell response to antigen varies with affinity, finding that association constant $K_a = 1/K_D$ values above $10^{10}M^{-1}$ did not lead to increased B cell triggering. This value is roughly equal to one derived previously (**Foote and Eisen, 1995**) starting from two assumptions: that diffusion places a fundamental upper limit to $k_{\text{on}}$ (where $K_D = k_{\text{off}}/k_{\text{on}}$), and that dissociation half-lives are less than one hour. Using our naive $K_D$ of $4x10^{-8}M$ (**Tas et al., 2016**), the ceiling $K_a$ value of $10^{10}M^{-1}$ corresponds to an affinity in our plots of $-log_{10}\frac{10^{-10}}{4x10^{-8}} = 2.6$, a value at which we observe few enough values in data (**Figure 4**) that our inferred response function there is reflective of our assumed sigmoid shape rather than any direct information from data. Also note that the additive, DMS-based affinity measurements that we use are only validated between about −1 and 2 (**DeWitt et al., 2025**, **Figure 2**).

A minimum, 'threshold' $K_a$ value of $10^6 M^{-1}$, below which B cells do not respond, has also been reported (**Batista and Neuberger, 1998**). While we do not attempt to model such a threshold in our simulation, this value would correspond to an affinity of $-\log_{10} \frac{10^{-6}}{4x10^{-8}} = -1.4$ in our plots, which is a region in which we observe the expected flat response function, although note that we also observe very few cells here (**Figure 4**).

## Direct inference of the affinity–fitness relationship

We have undertaken two other, independent efforts to infer the specific relationship between affinity and fitness, and it would be useful to compare their results. In **DeWitt et al., 2025**, we applied a traveling wave fitness approach (**Neher, 2013**; **Nourmohammad et al., 2013**) to the replay bulk data. Rather than, as in this paper, treating the evolution of individual cells and their BCR sequences, this models the distribution of cell fitnesses $p(x, t)$ as a traveling wave. The time evolution of $p(x, t)$ is controlled by two parameter functions: $q(x, y)$ describing the mutational flux from affinity state $x$ to state $y$ (derived from the DMS measurements), and the fitness landscape $f(x)$ specifying the fitness of a cell with affinity $x$ (an arbitrary function fit as part of the inference procedure). The net rate of population growth for a cell with affinity $x$ at time $t$ is thus $f(x) - \overline{f(t)}$, with $\overline{f(t)}$ the mean fitness of the population at time $t$. This difference can be thought of as an effective birth rate analogous to our $m\lambda - \mu$ (**Figure 7**). The evolution of $p(x, t)$ is then determined by the differential equation:

$$\frac{\partial}{\partial t}p(x, t) = \underbrace{\left(f(x) - \overline{f(t)}\right) p(x, t)}_{\text{net population growth}} + \underbrace{\int_{-\infty}^{\infty} (q(y, x)p(y, t) - q(x, y)p(x, t))dy}_{\text{net mutation to state } x}, \tag{7}$$

where we see that the dynamics of the traveling wave $p$ are given by the sum of two terms: net population growth and mutational flux.

We compared the traveling wave model's net growth rate to our effective birth rate in **Figure 8**, finding similar behavior for above-average affinities, but divergent values below. While this discrepancy is certainly noteworthy, it likely reflects a difference in input assumptions rather than truly different inferences. Our sigmoid model forces a horizontal asymptote in effective birth rate whose value is fixed by the relatively plentiful values near zero relative affinity. At the more negative values where the discrepancy becomes significant, the effective birth rate simply reflects our sigmoid assumption, with little input from the relatively sparse data. While we have not performed this analysis for our per-bin model, it also depends on a sigmoid (implicitly via training data), so would likely display similar behavior. The net growth rate from the traveling wave model, on the other hand, is more flexible.

Although our birth-death model and the traveling wave model have similar intent, they have very different structures and assumptions. The traveling wave represents the population as a continuous density $p(x, t)$ over affinity space, taking a large-population, deterministic limit. It also assumes a steady state with respect to total population size, meaning the distribution $p(x, t)$ evolves but total population size does not. In contrast, we simulate individual discrete cells, each with its own sequence, affinity, and stochastic fate. We explicitly model population dynamics: growth from a small founder population toward carrying capacity, with the possibility of stochastic extinction. This distinction is particularly important early in the GC reaction, when populations are small and growing. During this phase, stochastic effects are strongest, the extinction risk is highest, and the dynamics differ qualitatively from the later quasi-steady-state regime. It is remarkable that abstracting away so many granular biological details and simply modeling a two-dimensional probability distribution yields a coherent picture of GC evolution; however, it is far from clear that such simplification has no effect on the resulting inference. Indeed, in our model, we see clear effects of stochasticity, where single mutations on single cells can quickly sweep through the population, yielding very different trajectories for different GCs. The discrepancy at below-average affinities could also be related to differing treatments of expression. Our model completely ignores it, whereas real B cells must maintain a minimal expression level in order to survive.

The clear advantage of the traveling wave model is its simplicity: if its high-level view is accurate enough to effectively model the relevant GC dynamics, it is far more tractable. But reproducing low-level biological detail and making high-dimensional real data comparisons (e.g., **Figure 5**) to iteratively improve model fidelity are also useful, providing direct evidence that we are correctly modeling the underlying biological processes. The two approaches also utilize different types of data: we use a

single timepoint and thus must reconstruct evolutionary history, whereas the traveling wave requires a series of timepoints. The availability of both types of data is a unique feature of the replay experiment and provides us with the opportunity to directly compare the approaches.

We also note the different datasets used by the two methods. The bulk data used by the traveling wave model consists of counts of affinity values from bulk sequencing at seven timepoints (from 8 to 70 days), whereas our model uses phylogenetic trees with sequence and affinity annotations from single cell data extracted from GCs at two timepoints (15 and 20 days). Aside from the difference in available information between affinity counts and annotated trees, the longer time scale of the bulk data results in a larger range of measured affinity values. The bulk data also merges sequences from many GCs together before inference, whereas the extracted GC data analyzes each GC separately before jointly inferring results.

Both the intrinsic and effective birth rates are useful quantities, with each being appropriate for different tasks. The intrinsic rate is an unchanging, fundamental property of the GC system; but by itself, it does not tell us how many offspring to expect for a given cell in a particular GC. The effective rate, on the other hand, tells us the expected number of offspring for each cell at each point in time, but with the tradeoff that it has no fixed form: it depends, through $m$ (*Equation 2*), on the sum over cells of both birth and death rates, as well as on the number of cells alive at that time. Also note that because it depends on the particular population of cells alive at a given time, it is not defined along the entire $x$ axis, but only at $x$ values at which living cells reside (calculating its value at an affinity that no cells have would presuppose adding a cell at that affinity, which would modify the calculation for all other cells).

We have also worked to infer the response function using likelihoods under a birth-death model (*Bakis et al., 2025*). This requires solutions of ordinary differential equations (ODEs) that account for unsampled lineages in birth–death models. Because these models require independent evolution of lineages, they cannot incorporate the effects of either carrying capacity or competition between cells. They also require Markov type evolution, whereas sequence evolution in the GC is not Markovian. While we thus do not expect these likelihood-based results to describe real data, it is nevertheless instructive to compare them to our effective birth rates. The main difference is the much earlier ceiling in *Bakis et al., 2025*, plateauing entirely by around $x = 0.5$. This early plateau is simply a consequence of the lack of population size constraint: without a carrying capacity, the model can only avoid unbounded population growth with an early, and low, fitness ceiling. Since capacity constraint cannot be ignored in the context of the GC, indeed providing the foundation of selection there, this likelihood-based work is best understood as a substantial mathematical advance toward a potential future method that could infer response functions on real data.

Recent work has developed birth-death processes with mean-field interactions (*DeWitt et al., 2024*) that could be used for inference under models with population size constraint. Although the theoretical promise for such models is clear, inference has not yet been implemented. In future work, it would be interesting to compare these methods with those presented in this paper.

## Limitations

Our work is subject to a number of limitations, most importantly related to our model and simulation. As noted above, several aspects of our model's organization may not accurately reflect real GC dynamics. Furthermore, even for aspects that are accurately modeled, chosen parameter values may either not reflect true biological values or may represent subtly different 'effective' parameters and thus also need modification. Our simulation also does not include recent results showing that the mutation rate may be depressed after a strong light-zone signal (*Pae et al., 2025*), or that lower affinity cells may be subject to higher SHM rates (*Li et al., 2025*).

Other possible misspecifications involve dark-zone/light-zone cycling and cell sampling. Our simulation implements a single-generation proliferative phase with mutation (dark zone) always followed by a single, separate selection phase (light zone). However, this may not accurately represent real GC dynamics; for instance, this cannot simulate multiple proliferative phases between each selection phase ('clonal bursts') (*Tas et al., 2016*; *Pae et al., 2025*). It is also known that some selection occurs in the dark zone (*Mayer et al., 2017*), at least against nonfunctional cells, and our two-category death rate may not be sufficient to model this accurately. We also sample uniformly randomly from all living cells at the final timepoint, but it is possible that in real GCs, cell sampling is based on more

biased processes, for instance, if higher affinity cells, or cells in a particular zone, are more likely to be sampled.

We also neglect competition among lineages stemming from different rearrangement events (different clonal families), instead assuming that each GC is seeded with instances of only a single naive sequence, and that neither cells nor antibodies migrate between different GCs. More realistically for the polyclonal GC case, we would allow lineages stemming from different naive sequences to compete with each other both within and between GCs (*Zhang et al., 2013*; *McNamara et al., 2020*; *Barbulescu et al., 2025*). Implementing competition among several clonal families within a single GC would be conceptually simple and computationally practical in our current software framework. Competition among many GCs, however, would be computationally prohibitive because our time required is primarily determined by the total population size, since at each step we must iterate over every node and every event type in order to find the shortest waiting time. For the monoclonal replay experiment specifically, however, all naive sequences are the same, and so the current modeling framework is sufficient.

We also assume a single, constant death rate for functional cells. While we infer this rate with our summary statistic matching step, if the real rate depends on cell properties (other than simply being functional), the results will be incorrect. Because birth and death rates are interrelated mechanistically through phenomena such as tonic signaling (*Long et al., 2022*) and BCR expression-dependent birth rates (*Lam et al., 1997*), it is also unclear how much sensitivity we have to infer them independently. We note in passing that identifiability of birth and death rates from phylogenies is inherently difficult (*Morlon et al., 2022*).

It is also possible that our choice of a sigmoid for the response function is not sufficiently flexible. This form cannot, for instance, accommodate either a decrease in fitness with affinity or a discontinuous change in slope. It also requires symmetry around its midpoint. Because both the sigmoid and per-bin models were trained on sigmoid simulation, they can only infer sigmoid-like response functions.

While we chose to use a birth-modulated carrying capacity, it is possible that this does not accurately describe GC dynamics. We have, for instance, experimented with two other methods of capacity enforcement, one that modulates death, rather than birth, using the inverse of 2. The other, 'hard', constraint immediately kills a random individual whenever a birth event causes the population to exceed the carrying capacity.

Our summary statistic distributions are not always well-matched to data. The worst discrepancies are inaccurate structure (extra peaks) in the simulation affinity distributions, and deficits in the tails of the simulation abundance plots. The extra affinity peaks could be from inaccuracies in the sequence-affinity mapping, which assumes zero epistasis. Meanwhile, enlarging the abundance tail is easily accomplished by increasing burstiness with a steeper response function, or by decreasing diversity via either smaller initial population or smaller carrying capacity. These changes would also, of course, affect other distributions, with, for instance, a steeper response function increasing the speed of evolution and shifting the affinity distribution rightwards (while a smaller population does the opposite). While we can thus move the worst discrepancies from one distribution to another, our inability to match all distributions simultaneously is probably due either to one of the model misspecifications described in the preceding paragraphs, or to a neural network inference problem leading to incorrect response function.

Our neural network could also be improved. While the CBLV encoding incorporates a number of improvements over previous schemes, the process of encoding trees for digestion by neural networks is still a very new problem, and current methods are likely not particularly close to optimal. Recent work, for instance, has improved on the CBLV scheme by incorporating the generational context of each node in a way that makes it easier for the network to understand (*Perez and Gascuel, 2024*). This work also introduces a network architecture that is specifically designed around the new encoding. In the future, we plan to explore the use of this new method.

The replay experiment also represents a very special case among typical BCR repertoires. While the basic dynamics at play are likely to be representative of those in wild repertoires, it is important to keep in mind that these results only reflect mutation resulting from the exposure of a single naive antibody to a single experimental antigen. Some aspects of behavior in the low-SHM/early times regime of the extracted GC data are also potentially different from those at the higher SHM levels and longer

times found in typical repertoires. This is especially relevant to affinity or fitness ceilings, to which we likely have little sensitivity with the current data. Additionally, repeating the replay experiment using a less-optimized antigen with lower starting affinity might enable us to better map out dynamics during the early stages of affinity maturation when, for example, specificity may have some plasticity (*Zuo et al., 2025*).

## Conclusion and outlook

In closing, simulation-based methods have enabled inference on this otherwise-intractable, messy biological problem. However, they were not a panacea. Simulation-based inference is difficult when the model's behavior is determined by many parameters, since the product of many dimensions cannot be tested exhaustively. The optimality surface was also found to be quite rugged: it was easy to find combinations of parameters that lead to pathologies. Previous successful studies using simulation-based inference on trees have had fewer parameters and a less complex objective (*Voznica et al., 2022*; *Thompson et al., 2024*). Nevertheless, we were able to infer parameters of a complex and dynamic process that is hidden inside the germinal center.

Looking forward, even if by some miracle we could perform inference under an arbitrary forward-time model, interesting biological questions would remain. For instance, what mechanism limits the population size of the germinal center? In this paper, we chose birth-limited population size control, and the validity of our results likely depends somewhat on this assumption. In comparison with the traveling wave model of *DeWitt et al., 2025*, which has a separate structure and different assumptions, we could see that the inferred properties differed. Finding a convincing answer to this question will require additional experiments and likely additional modeling.

## Acknowledgements

We would like to thank all the authors of *DeWitt et al., 2025*, as well as the members of the Matsen lab for helpful discussion. This work is supported by NIH grants R01-AI146028 (PI Matsen), U01 AI150747 (PI Rustom Antia), R56-HG013117 (PI Song), and R01-HG013117 (PI Song). Frederick Matsen is an investigator of the Howard Hughes Medical Institute. Scientific Computing Infrastructure at Fred Hutch funded by ORIP grant S100D028685.

## Additional information

### Funding

| Funder | Grant reference number | Author |
|---|---|---|
| NIH Office of the Director | R01-AI146028 | Frederick A Matsen |
| NIH Office of the Director | U01 AI150747 | Frederick A Matsen |
| NIH Office of the Director | R56-HG013117 | Yun S Song |
| NIH Office of the Director | R01-HG013117 | Yun S Song |
| Howard Hughes Medical Institute | | Frederick A Matsen |

The funders had no role in study design, data collection and interpretation, or the decision to submit the work for publication.

### Author contributions

Duncan K Ralph, Conceptualization, Data curation, Software, Validation, Visualization, Writing – original draft, Writing – review and editing; Athanasios G Bakis, Conceptualization, Validation, Writing – review and editing; Jared G Galloway, Data curation, Software; Ashni A Vora, Tatsuya Araki, Data curation; Gabriel D Victora, Conceptualization, Data curation, Funding acquisition; Yun S Song, Conceptualization, Writing – review and editing; William S DeWitt, Conceptualization, Software, Writing – review and editing; Frederick A Matsen, Conceptualization, Software, Supervision, Funding acquisition, Writing – original draft, Writing – review and editing

Author ORCIDs
Duncan K Ralph (ID) https://orcid.org/0000-0002-2527-8610
Athanasios G Bakis (ID) https://orcid.org/0000-0002-5730-1483
Yun S Song (ID) https://orcid.org/0000-0002-0734-9868
Frederick A Matsen (ID) https://orcid.org/0000-0003-0607-6025

Reviewer #1 (Public review): https://doi.org/10.7554/eLife.108880.3.sa1
Reviewer #2 (Public review): https://doi.org/10.7554/eLife.108880.3.sa2
Author response https://doi.org/10.7554/eLife.108880.3.sa3

## Additional files

### Supplementary files
MDAR checklist

### Data availability
All raw and processed real and simulated data samples are available at: https://doi.org/10.5281/zenodo.15022130. All code is available at: https://github.com/matsengrp/gcdyn (copy archived at *Ralph et al., 2026*).

The following dataset was generated:

| Author(s) | Year | Dataset title | Dataset URL | Database and Identifier |
|---|---|---|---|---|
| Matsen FA, Ralph DK | 2025 | Inference of germinal center evolutionary dynamics via simulation-based deep learning | https://doi.org/10.5281/zenodo.15022130 | Zenodo, 10.5281/zenodo.15022130 |

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
