## [Editor Report · eLife Assessment]

This paper presents a computational method to infer from data a key feature of affinity maturation: the relationship between the affinity of B-cell receptors and their fitness. The approach, which is based on a simple population dynamics model but inferred using AI-powered simulation-based inference, is novel and **valuable**. It exploits recently published data on replay experiments of affinity maturation. The method is well argued and presented, and the validation is **compelling**.

---

## [Referee Report · Reviewer #1 (Public review)]

Summary:

This paper aims to characterize the relationship between affinity and fitness in the process of affinity maturation. To this end, the authors develop a model of germinal center reaction and a tailored statistical approach, building on recent advances in simulation-based inference.

The model provides a framework for linking affinity measurements and sequence evolution and does so while accounting for the stochasticity inherent to the germinal center reaction. The model's sophistication comes at the cost of numerous parameters and leads to intractable likelihood, which are the primary challenges addressed by the authors. The approach to inference is innovative and relies on training a neural network on extensive simulations of trajectories from the model.

The revised methods section is easier to follow and better explains the approach. Inference results on simulated data are compelling and the real-data findings are compared with alternative approaches, clarifying the relationship to previous work.

---

## [Referee Report · Reviewer #2 (Public review)]

Summary:

This paper presents a new approach for explicitly transforming B cell receptor affinity into evolutionary fitness in the germinal center. It demonstrates the feasibility of using likelihood-free inference to study this problem and demonstrates how effective birth rates appear to vary with affinity in real-world data.

Strengths:

The authors leverage the unique data they have generated for a separate project to provide novel insights to a fundamental question.The paper is clearly written, with accessible methods and straightforward discussion of the limits of this model.Code and data are publicly available and well-documented.

Weaknesses:

No substantial weaknesses noted.

---

## [Author Response]

The following is the authors’ response to the original reviews.

**Public Reviews:**

**Reviewer #1 (Public review):**
Summary:This paper aims to characterize the relationship between affinity and fitness in the process of affinity maturation. To this end, the authors develop a model of germinal center reaction and a tailored statistical approach, building on recent advances in simulation-based inference. The potential impact of this work is hindered by the poor organization of the manuscript. In crucial sections, the writing style and notations are unclear and difficult to follow.

We thank the reviewer for their kind words, and have endeavored to address all of their concerns as to the structure and style of the manuscript.

Strengths:The model provides a framework for linking affinity measurements and sequence evolution and does so while accounting for the stochasticity inherent to the germinal center reaction. The model's sophistication comes at the cost of numerous parameters and leads to intractable likelihood, which are the primary challenges addressed by the authors. The approach to inference is innovative and relies on training a neural network on extensive simulations of trajectories from the model.Weaknesses:The text is challenging to follow. The descriptions of the model and the inference procedure are fragmented and repetitive. In the introduction and the methods section, the same information is often provided multiple times, at different levels of detail.

Thank you for pointing this out. We have rearranged the methods in order to make the presentation more linear, and to reduce duplication with the introduction.

Specifically, we moved the affinity definition to the start, removed the redundant bullet point list, and moved the parameter value table to the end.

This organization sometimes requires the reader to move back and forth between subsections (there are multiple non-specific references to "above" and "below" in the text).

This is a great point, we have either removed or replaced all references to "above" or "below" with more specific citations.

The choice of some parameter values in simulations appears arbitrary and would benefit from more extensive justification. It remains unclear how the "significant uncertainty" associated with these parameters affects the results of inference.

We have clarified where various parameter values come from:

“In addition to the four sigmoid parameters, which we infer directly, there are other parameters in Table 1 about which we have incomplete information. The carrying capacity method and the choice of sigmoid for the response function represent fundamental model assumptions. We also fix the death rate for nonfunctional (stop) sequences, which would be very difficult to infer with the present experiment. For others, we know precise values from the replay experiment for each GC (time to sampling, # sampled cells/GC), but use a somewhat wider range for the sake of generalizability. The mutability multiplier is a heuristic factor used to match the SHM distributions to data. The naive birth rate is determined by the sigmoid parameters, but has its own range in order to facilitate efficient simulation.

For two of the three remaining parameters (carrying capacity and initial population), we can ostensibly choose values based on the replay experiment. These values carry significant uncertainty, however, partly due to inherent experimental uncertainty, but also because they may represent different biological quantities to those in simulation. For instance, an experimental measurement of the number of B cells in a germinal center might appear to correspond closely to simulation carrying capacity. However if germinal centers are not well mixed, such that competition occurs only among nearby cells, the "effective" carrying capacity that each cell experiences could be much smaller.

Fortunately, in addition to the neural network inference of sigmoid parameters, we have another source of information that we can use to infer non-sigmoid parameters: summary statistic distributions. We can use the matching of these distributions to effectively fit values for these additional unknown parameters. We also include the final parameter, the functional death rate, in these non-sigmoid inferred parameters, although it is unconstrained by the replay experiment, and it is unclear whether it is uniquely identifiable.”

In addition, the performance of the inference scheme on simulated data is difficult to evaluate, as the reported distributions of loss function values are not very informative.

We thought of two different interpretions for this comment, so have worked to address both.

First, the comment could have been that the distribution of loss functions on the training sample does not appear to be informative of performance on data-like samples. This is true, and in our revision we have emphasized the distinction between the two types of simulation sample: those for training, where each simulated GC has different (sampled) parameter values; vs the "data mimic" samples where all GCs have identical parameters. Since the former have different values for each GC, we can only plot many inferred curves together on the latter. We also would like to emphasize that the inference problem for one GC will have much more uncertainty than will that for an ensemble of GCs (as in the full replay experiment).

“After building and training our neural network, we evaluate its performance on subsets of the training sample. While this evaluation provides an important baseline and sanity check, it is important to note that the training sample differs dramatically from real data (and the “data mimic” simulation sample that mimics real data). While real data consists of 119 GCs with identical parameters and thus response functions, we need the GCs in our training sample to span the space of all plausible parameter values. This means that while we must evaluate performance on individual GCs in the training and testing samples, in real data (and data mimic simulation) we combine results from 119 curves into a central (medoid) curve. Inference on the training sample will thus appear vastly noisier than on real data and data mimic simulation, and also cannot be plotted with all true and inferred curves together.”

A second interpretation was that the reviewer did not have an intuitive sense of what a loss function value of, say, 1.0 actually means. To address this second interpretation, we have also added a supplement to Figure 2 with several example true and inferred response functions from the training sample, with representative loss values spanning 0.17 to 2.18. We have also added the following clarification to the caption of Figure 1-figure supplement 2:

“The loss value is thus the fraction of the area under the true curve represented by the area between the true and inferred curves.”

Finally, the discussion of the similarities and differences with an alternative approach to this inference problem, presented in Dewitt et al. (2025), is incomplete.

We have expanded this section of the manuscript, and added a new plot directly comparing the methods.

“In order to compare more directly to DeWitt et al. 2025, we remade their Fig.S6D, truncating to values at which affinities are actually observed in the bulk data, and using only three of the seven timepoints (11, 20, and 70, Figure 8, left). We then simulated 25 GCs with central data mimic parameters out to 70 days. For each such GC, we found the time point with mean affinity over living cells closest to each of three specific “target” affinity values (0.1, 1.0, 2.0) corresponding to the mean affinity of the bulk data at timepoints 11, 20, and 70. We then plot the effective birth rates of all living cells vs relative affinity (subtracting mean affinity) at the resulting GC-specific timepoints for all 25 GCs together Figure 8, right. Note that because each GC evolves at very different and time-dependent rates, we could not simply use the timepoints from the bulk data, since each GC slice from our simulation would then have very different mean affinity. The mean over GCs of these GC-specific chosen times is 10.9, 24.5, 44.4 (compared to the original bulk data time points 11, 20, 70). It is important to note that while the first two target affinities (0.1 and 1.0) are within the affinity ranges encountered in the extracted GC data, the third value (2.0) is far beyond them, and thus represents extrapolation to an affinity regime informed more by our underlying model than by the real data on which we fit it.”

**Reviewer #2 (Public review):**
Summary:This paper presents a new approach for explicitly transforming B-cell receptor affinity into evolutionary fitness in the germinal center. It demonstrates the feasibility of using likelihood-free inference to study this problem and demonstrates how effective birth rates appear to vary with affinity in real-world data.Strengths:(1) The authors leverage the unique data they have generated for a separate project to provide novel insights into a fundamental question. (2) The paper is clearly written, with accessible methods and a straightforward discussion of the limits of this model. (3) Code and data are publicly available and well documented.Weaknesses (minor):(1) Lines 444-446: I think that "affinity ceiling" and "fitness ceiling" should be considered independent concepts. The former, as the authors ably explain, is a physical limitation. This wouldn't necessarily correspond to a fitness ceiling, though, as Figure 7 shows. Conversely, the model developed here would allow for a fitness ceiling even if the physical limit doesn't exist.

Right, whoops, good point. We've rearranged the discussion to separate the concepts, for instance:

“While affinity and fitness ceilings are separate concepts, they are closely related. An affinity ceiling is a limit to affinity for a given antigen: there are no mutations that can improve affinity beyond this level. This would result in a truncated response function, undefined beyond the affinity ceiling. A fitness ceiling, on the other hand, is an upper asymptote on the response function. Such a ceiling would result in a limit on affinity for a germinal center reaction, since once cells are well into the upper asymptote of fitness they are no longer subject to selective pressure.”

(2) Lines 566-569: I would like to see this caveat fleshed out more and perhaps mentioned earlier in the paper. While relative affinity is far more important, it is not at all clear to me that absolute affinity can be totally ignored in modeling GC behavior.

This is a great point, we've added a mention of this where we introduce the replay experiment in the Methods:

“It is important to note that this is a much lower level than typical BCR repertoires, which average roughly 5-10% nucleotide shm.”

And expanded on the explanation in the Discussion:

“Some aspects of behavior in the low-shm/early times regime of the extracted GC data are also potentially different to those at the higher shm levels and longer times found in typical repertoires. This is especially relevant to affinity or fitness ceilings, to which we likely have little sensitivity with the current data.”

(3) One other limitation that is worth mentioning, though beyond the scope of the current work to fully address: the evolution of the repertoire is also strongly shaped by competition from circulating antibodies. (Eg: http://www.ncbi.nlm.nih.gov/pmc/articles/PMC3600904/, http://www.sciencedirect.com/science/article/pii/S1931312820303978). This is irrelevant for the replay experiment modeled here, but still an important factor in general repertoires.

Yes good point, we've added these citations in a new paragraph on between-lineage competition:

“We also neglect competition among lineages stemming from different rearrangement events (different clonal families), instead assuming that each GC is seeded with instances of only a single naive sequence, and that neither cells nor antibodies migrate between different GCs. More realistically for the polyclonal GC case, we would allow lineages stemming from different naive sequences to compete with each other both within and between GCs (Zhang et al. 2013: McNamara et al. 2020; Barbulescu et al. 2025). Implementing competition among several clonal families within a single GC would be conceptually simple and computationally practical in our current software framework. Competition among many GCs, however, would be computationally prohibitive because our time required is primarily determined by the total population size, since at each step we must iterate over every node and every event type in order to find the shortest waiting time. For the monoclonal replay experiment specifically, however, all naive sequences are the same and so the current modeling framework is sufficient.”

**Recommendations for the authors:**

**Reviewing Editor Comments:**
The authors are encouraged to follow the suggestions of manuscript re-organization by Reviewer 1, in order to improve readability. We would also like to suggest improving the discussion of the traveling wave model to explain it in a more self-contained way. In passing, please clarify what is meant by 'steady-state' in that model. A superficial understanding would suggest that the only steady state in that model would be a homogeneous population of antibodies with maximum affinity/fitness.

These are great suggestions. We have substantially rearranged the text according to Reviewer 1's suggestions, especially the Methods, and expanded on and rearranged the traveling wave discussion. We've also clarified throughout that the traveling wave model is assuming steady state with respect to population. In the public response to reviewer 1 above we describe these changes in more detail.

**Reviewer #1 (Recommendations for the authors):**
I suggest that the organization of the paper be reconsidered. The current methods section is long and at times repetitive, making it impossible to parse in a single reading. Moving some technical details from the main text to an appendix could improve readability. Despite the length of the methods section, many important points, such as justification of choices in model specification or values of parameters, are treated only briefly.

We have rearranged the methods section, particularly the discussion of our model, and have more clearly justified choices of parameter values as described in the public response.

Discussion of similarities and differences with reference to Dewitt et al. 2025 should be revised, as it's currently unclear whether the method presented here has any advantages.

We have expanded this comparison, and emphasized the main disadvantage of the traveling wave approach: there is no way of knowing whether by abstracting away so much biological detail it misses important effects. We have also emphasized that the two approaches use different types of data (time series vs endpoint) which are typically not simultaneously available:

“The clear advantage of the traveling wave model is its simplicity: if its high level view is accurate enough to effectively model the relevant GC dynamics, it is far more tractable. But reproducing low-level biological detail, and making high-dimensional real data comparisons (e.g. Figure 5) to iteratively improve model fidelity, are also useful, providing direct evidence that we are correctly modeling the underlying biological processes. The two approaches also utilize different types of data: we use a single time point, and thus must reconstruct evolutionary history; whereas the traveling wave requires a series of timepoints. The availability of both types of data is a unique feature of the replay experiment, and provides us with the opportunity to directly compare the approaches.”

The results obtained from the same data should be directly compared (can the response function be directly compared to the result in Figure S6D in Dewitt et al., 2025? If yes, it should be re-plotted here and compared/superimposed with Figures 6 and 7). The text mentions the results differ, but it remains ambiguous whether the differences are significant and what their implications are.

We've added a new Figure 8, comparing a modified version of the traveling wave Fig S6D to a new plot derived from our results using the data mimic parameters. While the two plots represent fundamentally different quantities, they do put the results of the two methods on an approximately equal footing and we see nice concordance between them in regions with significant data (they disagree substantially for larger negative affinities). We have also added emphasis to the point that the traveling wave model uses an entirely separate dataset to what we use here.

Other comments:(1) l. 80: "[in] around 10 days"?

Text rearranged so this phrase no longer appears.

(2) l. 96: "an intrinsic rate [given by?] the response function above".

Text rearranged so this phrase no longer appears.

(3) Figure 1: The. “specific model” could part be expanded and improved to help make sense of model parameters and the order of different processes in the population model. Example values of parameters can be plotted rather than loosely described, e.g., y_h+y_c, the upper asymptotes can be plotted in place of the “yscale determines upper asymptotes” label.

Great suggestion, we've changed the labels.

(4) The cartoons in the other parts are somewhat cryptic or illegible due to small sizes.

We have added text in the caption linking to the figures that are, in the figure, intended to be in schematic form only.

“Plots from elsewhere in the manuscript are rendered in schematic form: those in “infer on data” refer to Figure 4-figure supplement 1, and those in “simulate with inferred parameters” to Figure 5.

(5) L. 137: It's not helpful to give numerical values before the definition of affinity. (and these numbers are repeated later).

Good point, we've moved the affinity definition to the previous section, and remove the duplicate range information.

(6): Table 1: A number of notations are unclear, such as “#seqs/GC” or “mutability multiplier”. The double notation for crucial parameters doesn't help. At the moment the table is introduced, the columns make little sense to the reader, and it's not well specified what dictates the choice or changes of parameter values or ranges.

We've moved the table further down until after the parameters have been introduced, and clarified the indicated names.

(7) l. 147: Choices of model are not justified and appear arbitrary (e.g., why death events happen at one of two rate).

We have clarified the reasoning behind having two death rates.

(8) l.151: “happened on the edges of developing phylogenetic tree” - ambiguous: do they accumulate at cell divisions? What is a “developing tree”?

We have removed this ambiguous phrasing.

(9) l.161: This paragraph is particularly dense.

We have rearranged this section of the methods, and split up this paragraph.

(10) l. 164: All the different response functions for different event types? Or only the one for birth, as stated before?

Yes. This has been clarified.

(11) l.167: Does the statement in the bracket refer to a unit?

This has been clarified.

(12) l. 169: Discussion of the implementation seems too detailed.

Hopefully the rearranged description is clearer, but we worry that removing the details of events selection would leave some readers confused.

(13) l. 186: Why describe the methods that, in the end, were not used? Similarly, as a mention of “variety of response functions” seems out of place if only one choice is used throughout the paper. eq. (2): that's mˆ{-1} from eq. (1). Having the two equations using the same notation is confusing.

We've moved the mention of alternatives to the Discussion, where it is an important source of uncontrolled systematic uncertainty, and removed the extra equation.

(14) l. 206: Unclear what “thus” refers to.

Removed.

(15) l.211: What does “neglecting y_h” mean?

This has been clarified.

(16) l. 242: Unclear what “this” refers to.

Clarified.

(17) l. 261: What does “model independence” refer to in this context?

From the sigmoid model. Clarified.

(18) l. 306: What values for which parameters? References?

We have clarified and updated this statement - it was out of date, corresponding to the analysis before we started fitting non-sigmoid parameters.

“In addition to the four sigmoid parameters, which we infer directly, there are other parameters in Table 1 about which we have incomplete information. The carrying capacity method and the choice of sigmoid for the response function represent fundamental model assumptions. We also fix the death rate for nonfunctional (stop) sequences, which would be very difficult to infer with the present experiment. For others, we know precise values from the replay experiment for each GC (time to sampling, # sampled cells/GC), but use a somewhat wider range for the sake of generalizability. The mutability multiplier is a heuristic factor used to match the SHM distributions to data. The naive birth rate is determined by the sigmoid parameters, but has its own range in order to facilitate efficient simulation.

For two of the three remaining parameters (carrying capacity and initial population), we can ostensibly choose values based on the replay experiment. These values carry significant uncertainty, however, partly due to inherent experimental uncertainty, but also because they may represent different biological quantities to those in simulation. For instance, an experimental measurement of the number of B cells in a germinal center might appear to correspond closely to simulation carrying capacity. However if germinal centers are not well mixed, such that competition occurs only among nearby cells, the "effective" carrying capacity that each cell experiences could be much smaller.

Fortunately, in addition to the neural network inference of sigmoid parameters, we have another source of information that we can use to infer non-sigmoid parameters: summary statistic distributions. We can use the matching of these distributions to effectively fit values for these additional unknown parameters. We also include the final parameter, the functional death rate, in these non-sigmoid inferred parameters, although it is unconstrained by the replay experiment, and it is unclear whether it is uniquely identifiable.”

(19) l. 326: "is interpreted as having" or "corresponds to"?

Changed.

(20) l. 340: Not sure what "encompassing" means in this context.

Clarified.

(21) l. 341: "We do this..." -- I think this sentence is not grammatical.

Fixed.

(22) l. 348: "on simulation" -- "from simulated data"?

Indeed.

(23) l. 351: "top rows", the figures only have one row.

Fixed.

(24) Figure 2: It's difficult to tell from the loss function itself whether inference on simulated data works well. Why not report the simulated and inferred response functions? The equivalent plots in Figure 5 would also be informative. Has inference been tested for different "sigmoid parameters" values?

This is an important point that was not clear, thanks for bringing it up. We have expanded on and emphasized the differences between these samples and the reasoning behind their different evaluation choices. Briefly, we can't display true vs inferred response functions on the training samples since the curves for each GC are different -- the plot would be entirely filled in with very different response function shapes. This is why we do actual performance evaluation on the "data mimic" samples, where all GCs have the same parameters. Summary stats (like Fig 5) for the training sample are in Fig 5 Supplement 2.

(25) l. 354: Unclear what "this" refers to.

Removed.

(26) l. 355: We assume the parameters are the same?

Yes, we assume all data GCs have the same parameters. We have added emphasis of this point.

(27) Figure 4: Is "lambda" the fitness? Should be typeset as \lambda_i?

Our convention is to add the subscript when evaluating fitness on individual cells, but to omit it, as here, when plotting the response function as a whole.

(28) l. 412: "[a] carrying capacity constraint".

Fixed.

**Reviewer #2 (Recommendations for the authors):**
(1) In 2 places, you state that observed affinity ranged from -37 to 3, but I assume that the lower bound should be -3.7.

The -37 was actually correct, but we had mistakenly missed updating it when we switched to the latest (current) version of the affinity model. We have updated the values, although these don't really have any effect on the model since we only infer within bounds in which we have a lot of points:

“Affinity is ∅ for the initial unmutated sequence, and ranges from -12.2 to 3.5 in observed sequences, with a mean median of -0.3 (0.3).

(2). I had to look up the Vols nicker paper to understand the tree encoding: It would be nice to spend another sentence or two on it here for those who aren't familiar.

Great point, we have added the following:

“We encode each tree with an approach similar to Lambert et al. (2023) and Thompson et al. (2024), most closely following the compact bijective ladderized vector (CBLV) approach from Voznica et al. (2022). The CBLV method first ladderizes the tree by rotating each subtree such that, roughly speaking, longer branches end up toward the left. This does not modify the tree, but rather allows iteration over nodes in a defined, repeatable way, called inorder iteration. To generate the matrix, we traverse the ladderized tree in order, calculating a distance to associate with each node. For internal nodes, this is the distance to root, whereas for leaf nodes it is the distance to the most-recently-visited internal node (Voznica et al., 2022, Fig. 2). Distances corresponding to leaf nodes are arranged in the first row of the matrix, while those from internal nodes form the second row.”

(3) On line 351, you refer to the "top rows of Figure 2 and Figure 3," but each only has one row in the current version. I think it should now be "left panel.".

Fixed.

(4) How many vertical dashed lines are in the left panel of the bottom row of Figure 7? I think it's more than one, but can't tell if it is two or three...

Nice catch! There were actually three. We've shortened them and added a white outline to clarify overlapping lines.

(5) Would the model be applicable to GCs with multiple naive founders of different affinities? Or would more/different parameters be needed to account for that?

The model would be applicable, but since the time required for our simulation scales roughly with the total simulated population size, we could probably only handle competition among at most a couple of GCs. Some sort of "migration strength" parameter would be required for competition among GCs (or within one GC if we don't want to assume it's well-mixed), but that doesn't seem a terrible impediment. We've added the following:

“We also neglect competition among lineages stemming from different rearrangement events (different clonal families), instead assuming that each GC is seeded with instances of only a single naive sequence, and that neither cells nor antibodies migrate between different GCs. More realistically for the polyclonal GC case, we would allow lineages stemming from different naive sequences to compete with each other both within and between GCs (Zhang et al. 2013; McNamara et al. 2020; Barbulescu et al. 2025). Implementing competition among several clonal families within a single GC would be conceptually simple and computationally practical in our current software framework. Competition among many GCs, however, would be computationally prohibitive because our time required is primarily determined by the total population size, since at each step we must iterate over every node and every event type in order to find the shortest waiting time. For the monoclonal replay experiment specifically, however, all naive sequences are the same and so the current modeling framework is sufficient.”